# Severe 21st-century ocean acidification in Antarctic Marine Protected Areas

Cara Nissen [1,2] ✉, Nicole S. Lovenduski [1], Cassandra M. Brooks[3], Mario Hoppema [2], Ralph Timmermann[2] & Judith Hauck [2]

Antarctic coastal waters are home to several established or proposed Marine Protected Areas (MPAs) supporting exceptional biodiversity. Despite being threatened by anthropogenic climate change, uncertainties remain surrounding the future ocean acidification (OA) of these waters. Here we present 21st-century projections of OA in Antarctic MPAs under four emission scenarios using a high-resolution ocean–sea ice–biogeochemistry model with realistic ice-shelf geometry. By 2100, we project pH declines of up to 0.36 (total scale) for the top 200 m. Vigorous vertical mixing of anthropogenic carbon produces severe OA throughout the water column in coastal waters of proposed and existing MPAs. Consequently, end-of-century aragonite undersaturation is ubiquitous under the three highest emission scenarios. Given the cumulative threat to marine ecosystems by environmental change and activities such as fishing, our findings call for strong emission-mitigation efforts and further management strategies to reduce pressures on ecosystems, such as the continuation and expansion of Antarctic MPAs.

Ocean acidification (OA) is an environmental problem[1] threatening marine biodiversity across the global oceans[2–4]. OA is caused by the invasion of anthropogenic carbon ($C_{anth}$) into the ocean, which results in a decline in pH and in the saturation states of the calcium carbonate minerals aragonite ($\Omega_{arag}$) and calcite ($\Omega_{calc}$). The Southern Ocean is especially sensitive to OA due to the prevalence of high $CO_2$ concentrations in cold waters, the upwelling of carbon-rich deep waters with low pH values and a high Revelle factor[1,2,5]. Earth system model projections suggest that surface waters in the open Southern Ocean might become seasonally undersaturated with respect to the more soluble mineral aragonite ($\Omega_{arag} < 1$) by 2035[2,6–8]. Projections of OA on the Antarctic continental shelves have so far been elusive due to the inability of coarse-resolution models to realistically resolve physical and biogeochemical processes at high resolution and in ice-shelf cavities. Observations[9–24] and regional models[25,26] indicate that Antarctic shelf waters are a present-day sink for anthropogenic carbon ($C_{anth}$). This net sink is the result of strong biologically driven uptake in summer, while sea-ice cover typically prevents outgassing to the

atmosphere in winter[25]. Shelf waters are especially prone to OA resulting from rising atmospheric $CO_2$, reductions in sea-ice cover and enhanced freshwater input. Dense-water formation on these shelves implies that shelf OA signals can subsequently spread into the global abyssal ocean[11,15,16,20,21,23].

Designed to protect the unique high-latitude Southern Ocean biodiversity[27,28], a network of MPAs is being developed by the Commission for the Conservation of Antarctic Marine Living Resources (CCAMLR). MPAs have been established at the South Orkney Islands Southern Shelf and in the Ross Sea region, with three additional MPAs proposed in the Weddell Sea, East Antarctica and along the western Antarctic Peninsula (Fig. 1)[29–31]. If realized, this network of MPAs would protect ~60% of Antarctic shelf waters (outside of ice-shelf cavities). Key organisms in the high-latitude Southern Ocean ecosystem include primary producers such as diatoms and the haptophyte *Phaeocystis*[32,33], mid-trophic level organisms such as Antarctic krill, foraminifera, pteropods and smaller fish (e.g., Antarctic silverfish)[30,34–37] and upper-trophic level predators such as large fish

[1]Department of Atmospheric and Oceanic Sciences and Institute of Arctic and Alpine Research, University of Colorado Boulder, Boulder, CO, USA. [2]Alfred Wegener Institut, Helmholtz-Zentrum für Polar- und Meeresforschung, Bremerhaven, Germany. [3]Department of Environmental Studies and Institute of Arctic and Alpine Research, University of Colorado Boulder, Boulder, CO, USA. ✉e-mail: cara.nissen@colorado.edu

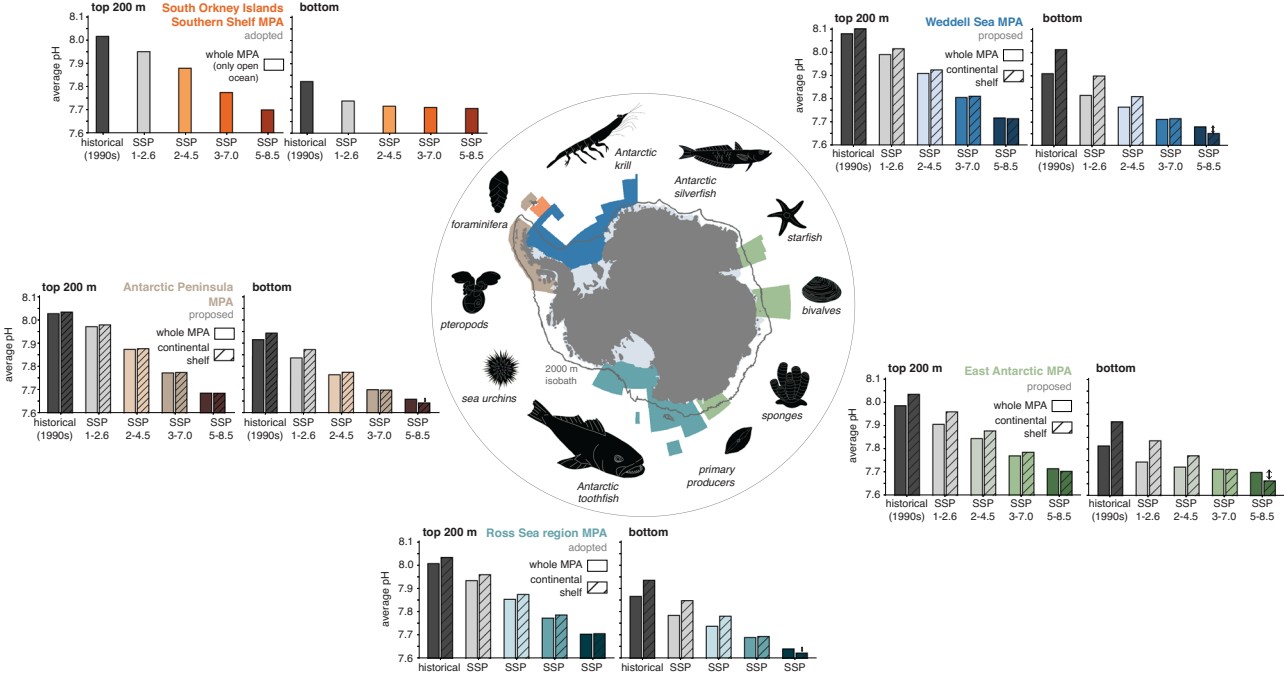

**Fig. 1 | Key organisms in the Southern Ocean and twenty-first-century ocean acidification in Marine Protected Areas.** Map of established (South Orkney Islands Southern Shelf and Ross Sea region) and proposed (Weddell Sea, East Antarctic, Western Antarctic Peninsula) Marine Protected Areas (MPAs). Key species and groups that may be negatively impacted by ocean acidification (see references in the main text) are depicted around the map (not to scale). Bar plots denote the top 200 m average (left) and bottom (right) pH in the five different MPAs in the 1990s (dark gray) and in the 2090s (colors) for the four emission scenarios (sorted from low emission to high emission). For the Weddell Sea, East Antarctic, Ross Sea and Antarctic Peninsula MPAs, the hatched bars denote the average pH for the continental shelf south of the 2000 m isobath (gray contour on the map). Note the arrows for the SSP5-8.5 scenario, indicating a lower pH on the continental shelves than for the whole-MPA average.

(e.g., Antarctic toothfish), sea birds, seals, whales and penguins[31,38–41]. In addition, the benthic ecosystem is highly diverse, including sponges, bivalves and echinoderms (e.g., starfish and sea urchins; see Fig. 1)[42–44]. New species and habitats, such as icefish nesting grounds, continue to be discovered in remote Antarctic coastal waters[45,46]. By preserving genetic, species and ecosystem diversity through limitation or prohibition of human activities (e.g., fishing), the implementation of MPAs can increase resilience of marine ecosystems to environmental change, including those resulting from climate change[47–49].

OA is one of many Southern Ocean ecosystem stressors arising from climate change, along with warming, deoxygenation and changes in sea-ice cover and circulation. As such, a change in each of these factors individually or in a multitude of factors combined directly impacts the fitness of marine organisms and ecosystem functioning[29,50–52]. A large number of studies suggest that physiological processes are negatively impacted by OA, with the severity of the impact varying across organisms and life stages. Due to a higher metabolic demand under OA, phytoplankton gross primary production is reduced at $pCO_2 > 800$ µatm, though species-specific differences, species interactions and potential adaptation are not yet well understood[3,53–60]. An >80% reduction in egg hatch rates and thus recruitment of Antarctic krill has been reported at $pCO_2 > 1500$ µatm[61], while adult krill appear to be resilient to elevated $pCO_2$[62,63]. Calcium carbonate shell dissolution in undersaturated waters ($\Omega < 1$) has been reported for benthic organisms[4], foraminifera[4] and pteropods[64,65]. Juvenile pteropods exhibit increased resting metabolic rates at $pCO_2 > 900$ µatm[66] and egg development is delayed at pH < 7.8[67]. In contrast, the development of juvenile dragonfish accelerates at pH < 7.8, but their mortality is highest when exposed to both OA and warming[68]. Juvenile Antarctic emerald rockcod and black rockcod show reduced metabolic capacity to adapt to warming when exposed

to pH < 7.75[69,70]. Benthic bivalve and sea urchin larvae are increasingly malformed and damaged a pH < 7.8, possibly compromising their survival[4,71]. Altogether, acknowledging knowledge gaps on direct OA impacts for important species in the Southern Ocean food web (e.g., Antarctic silverfish and upper-trophic level organisms), even small impacts of OA on individual species can accumulate through the food web, as shifts in development can cause predator-prey mismatches[62,67,68]. While the overall number of available studies is low and inter-species variability is potentially high[3,55,72], OA is among the cumulative threats to marine food webs that along with other direct threats, such as potential overfishing, stand to disrupt marine food webs in these sensitive Antarctic waters.

Here, we use a global ocean–sea ice–biogeochemistry model with eddy-permitting grid resolution on the Antarctic continental shelves and a representation of ice-shelf cavities to project twenty-first-century OA in the regions encompassed by the five adopted and proposed Antarctic MPAs under four "shared socioeconomic pathways" (SSP) emission scenarios[73,74]. The four SSP scenarios represent different pathways of societal development, ranging from SSP1 assuming an increasing importance of sustainable practices to the fossil-fuel intensive SSP5[75]. By the year 2100, the atmospheric forcing applied in this study corresponds to an increase in global 2 m air temperatures relative to the 1990s of 1.2 °C (SSP1-2.6), 2.3 °C (SSP2-4.5), 3.4 °C (SSP3-7.0) and 4.2 °C (SSP5-8.5)[76]. Atmospheric $CO_2$ levels in these scenarios will have increased to 446, 603, 867, and 1135 ppm[75] by the year 2100, respectively. Our results suggest a dramatic increase in the volume of calcium carbonate undersaturated MPA waters by 2100. Furthermore, we show that $C_{anth}$ mixes into shelf waters more readily than open-ocean waters, resulting in more severe OA on the continental shelves, where OA-sensitive ecosystems reside[4,44].

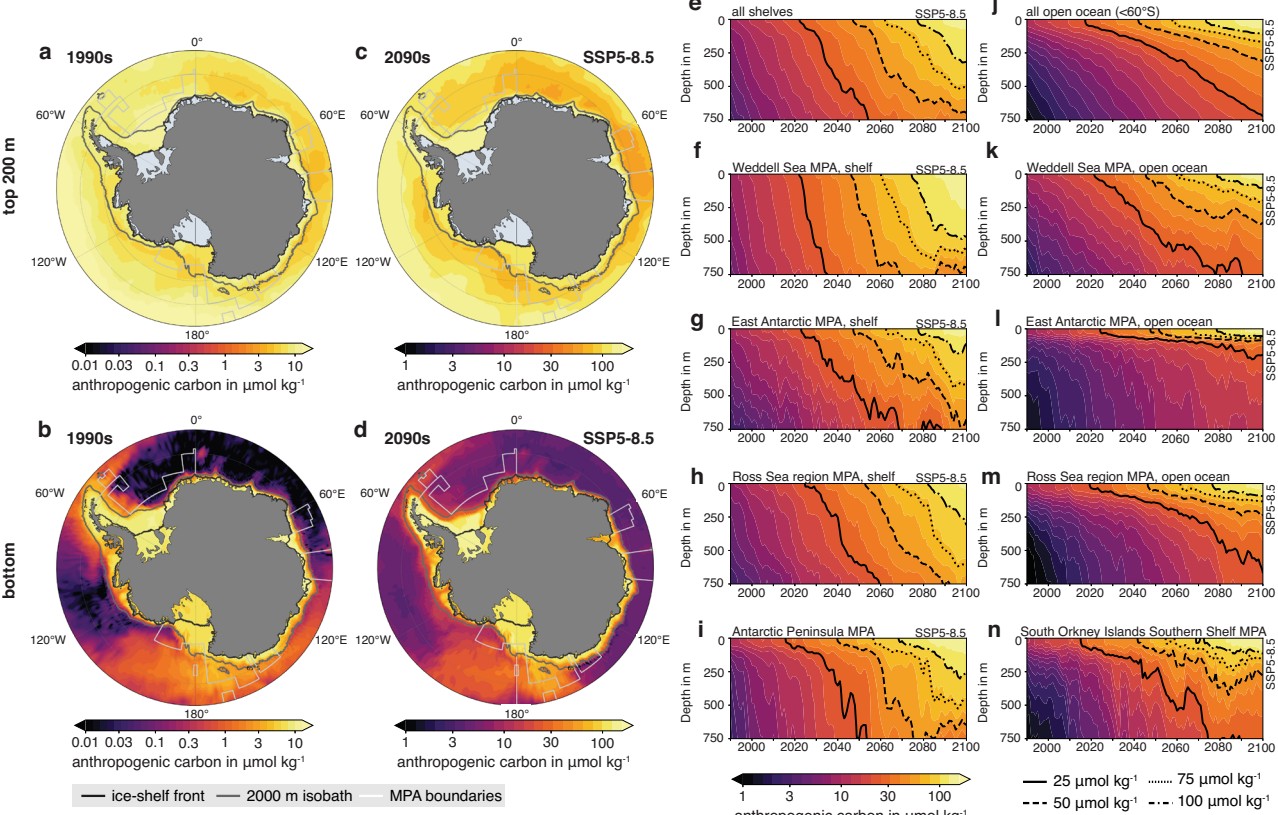

**Fig. 2 | Distribution of anthropogenic carbon across the high-latitude Southern Ocean.** Distribution of anthropogenic carbon concentrations in µmol kg[−1] **a** averaged over the top 200 m and **b** at the bottom in the 1990s of the historical simulation. **c**, **d** Same as (**a**) and (**b**), but for the 2090s of the high-emission scenario SSP5-8.5. The gray contour denotes the 2000 m isobath, which separates the continental shelf from the open ocean in this study. Note the logarithmic colorbar and the one order of magnitude difference between the 1990s and the 2090s. Outlines of the Marine Protected Areas (MPAs) are denoted in white and the ice-shelf front is shown as the dark gray contour. **e–h** Temporal evolution of anthropogenic carbon concentrations with depth and averaged over **e** all continental shelves (including areas outside of the MPAs) and the continental shelves of **f** the proposed Weddell Sea MPA, **g** the proposed East Antarctic MPA and **h** the adopted Ross Sea region MPA. **j–m** Same as (**e**)–(**h**), but for the open ocean. **i** and **n** show the averages for **i** the proposed MPA along the western Antarctic Peninsula and **n** the adopted South Orkney Islands Southern Shelf MPA.

## Results

### More severe ocean acidification on continental shelves than in the open ocean under high-emission scenarios

Simulated $C_{anth}$ concentrations averaged over the top 200 m of the water column are similar in magnitude on the continental shelves and in the open ocean under present-day conditions throughout the Antarctic (Fig. 2a). In contrast, future top 200 m $C_{anth}$ concentrations and those at the seafloor are at least two times higher on the continental shelves than in the open ocean for the high-emission scenario SSP5-8.5 (Fig. 2b–d). In the open ocean, bottom $C_{anth}$ concentrations are elevated in the main dense-water formation regions in the Weddell Sea and the Ross Sea and downstream of these regions (up to 4 and 70 µmol kg[−1] in some places for the 1990s and 2090s, respectively), whereas $C_{anth}$ concentrations on the continental shelves exceed 10 and 120 µmol kg[−1] in many places around the Antarctic continent (Fig. 2b, d). This shelf–open ocean gradient is in general agreement with observations, which indicate present-day shelf $C_{anth}$ concentrations of up to 50 µmol kg[−1][,16,21,22,24] (observations taken between 2003 and 2014) and bottom concentrations of up to 25 µmol kg[−1] in the Ross Sea region MPA[16,21,24] (2003–2014) and 16 µmol kg[−1] in the Weddell Sea MPA[9,17,18] (1988–2008).

The difference in average $C_{anth}$ concentrations between the continental shelves and the open ocean is caused by enhanced vertical mixing of $C_{anth}$ in Antarctic shelf waters[77], as illustrated by steeper slopes of the $C_{anth}$ isolines in Fig. 2e–i (continental shelves) than in Fig. 2j–n (open ocean). This enhanced vertical mixing is a signature of dense-water formation on the continental shelves, which readily transports $C_{anth}$ from the surface to depth[11,15,16,20,21,23]. Consequently, by 2100, $C_{anth}$ concentrations in the top 750 m exceed 50 µmol kg[−1] on all continental shelves in the high-emission scenario, while waters with such high concentrations are restricted to the top ~200 m in the open ocean. Given that the South Orkney Islands Southern Shelf MPA only encompasses waters that are deeper than 2000 m, it is unsurprising that $C_{anth}$ penetration in this region is confined to shallower depths than in all other MPAs. The strong vertical mixing on the shelves (where most of the MPA area is located, Fig. 1) is the ultimate cause of severe OA on continental shelves and in the MPAs studied.

In the 1990s, waters on the continental shelves had a higher pH compared to the average over the regions encompassed by the whole MPA (dashed black lines on top and to the right of the solid black lines in Figs. 3 and 4, respectively). Averaged over the top 200 m, the pH in Antarctic coastal waters was up to 0.05 higher in the 1990s than in the regions encompassed by the whole MPAs (with a maximum for the East Antarctic MPA), while this difference amounts to up to 0.1 for bottom pH (Weddell Sea and East Antarctic MPA; Fig. 1). Over the twenty-first century, this shelf–open ocean gradient is sustained for the lower-emission scenarios, but it reverses for the two highest-emission scenarios in the regions encompassed by the Weddell Sea, East Antarctic and Ross Sea region MPAs (dashed colored lines below and to the left

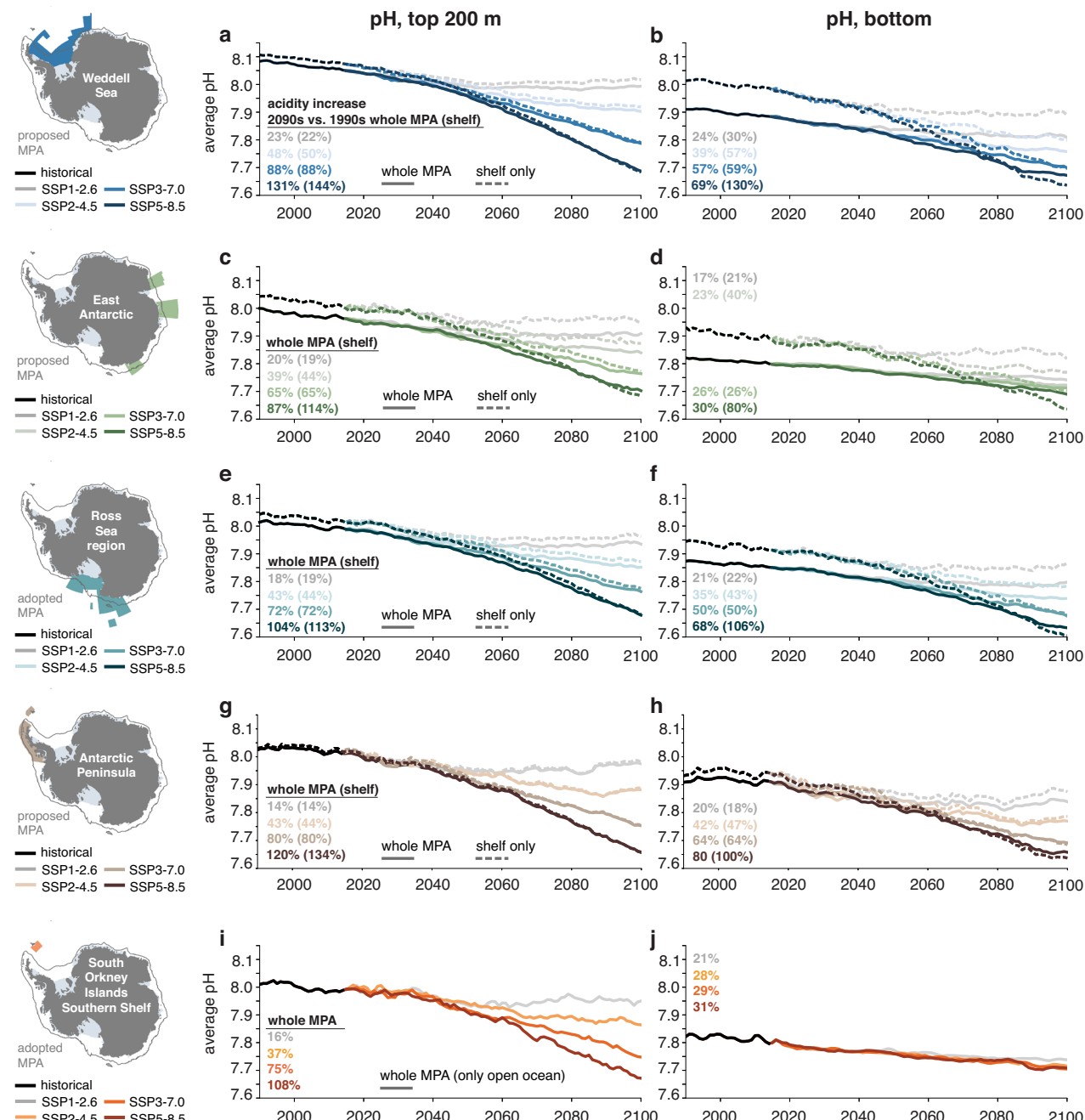

**Fig. 3 | Temporal evolution of pH. a, b** Time series of **a** the top 200 m average pH and **b** bottom pH in the proposed Weddell Sea Marine Protected Area (MPA) for the historical time period (black) and the SSP1-2.6 scenario (light gray), the SSP2-4.5 scenario (light blue), the SSP3-7.0 scenario (intermediate blue) and the SSP5-8.5 scenario (dark blue). Solid lines show the time series for the whole MPA and dashed lines for the continental shelves south of the 2000 m isobath (see dark gray contour in map). The numbers printed in each panel denote the percent increase in acidity, which was computed from the H⁺ concentration in the 2090s and the 1990s. **c–j** Same as (**a**) and (**b**), but for (**c**) and (**d**) the proposed East Antarctic MPA, **e, f** the adopted Ross Sea region MPA, **g, h** the proposed Antarctic Peninsula MPA, **i, j** the adopted South Orkney Islands Southern Shelf MPA. Note that only the time series for the whole MPA are shown in (**i**) and (**j**) (see Methods).

of the solid colored lines in Figs. 3 and 4, respectively). By 2100, the average pH on the continental shelves is up to 0.01 (top 200 m) and 0.04 (bottom) lower than in the area of the whole MPA, again demonstrating the overall stronger acidification in the shallow continental shelf seas than in the open ocean (Fig. 1; note that the region of the South Orkney Islands Southern Shelf MPA is exclusively "open ocean"; see Methods). Expressed as an increase in hydrogen ion concentration by the 2090s relative to the 1990s, the acidity on the continental shelves increases by up to 144% (top 200 m; Weddell Sea) and 130% (bottom; Weddell Sea), which is substantially more than for the

whole-MPA averages (up to 131% and 80%; see numbers printed into panels in Fig. 3).

Over the twenty-first century, both the magnitude and sign of the vertical pH gradient in the adopted and proposed MPAs are altered due to OA. Surface waters in all MPAs have a higher pH than waters at depth in the 1990s, with the pH difference between the surface and 1500 m ranging from 0.23 in the Ross Sea region MPA to 0.27 in the South Orkney Islands Southern Shelf MPA (black lines in Fig. 4). Yet, by the 2090s, surface waters are more acidic than waters at depth for the two highest-emission scenarios, with a reversed vertical pH gradient

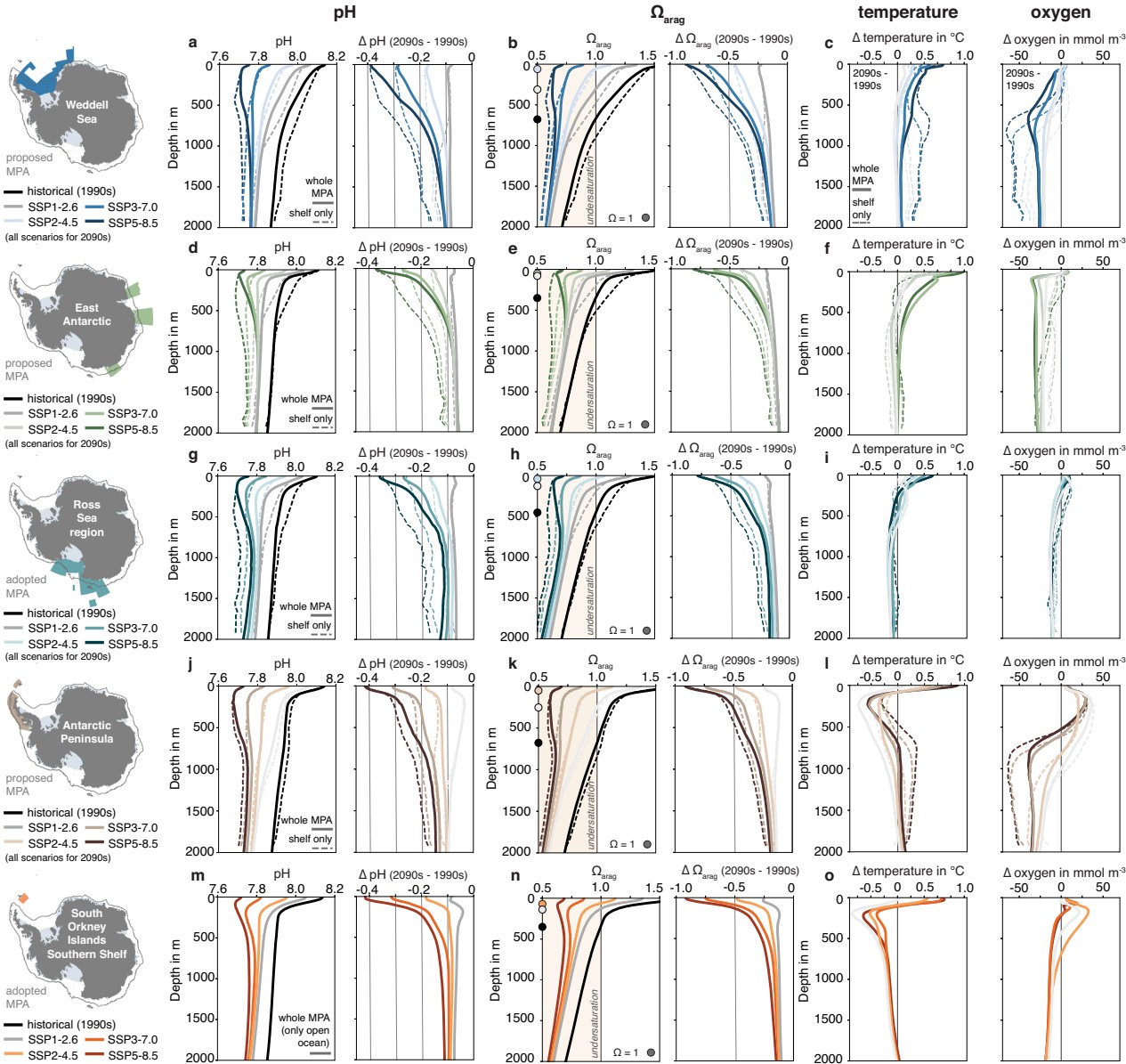

**Fig. 4 | Changes in the vertical distribution of pH, $\Omega_{arag}$, temperature and oxygen.** Vertical profiles of **a** pH and **b** the saturation state with respect to aragonite ($\Omega_{arag}$) in the proposed Weddell Sea Marine Protected Area (MPA). Solid lines show the profiles above 2000 m in the whole MPA and dashed lines show the profile for the continental shelves south of the 2000 m isobath (see dark gray contour in map). The black lines denote the 1990s in the historical simulation and the colored lines the 2090s in the four emission scenarios. The profiles on the left in each panel show absolute values and profiles on the right show the change in each property between the 2090s and the 1990s (ΔpH and $\Delta\Omega_{arag}$). In **b**, undersaturated conditions, i.e., $\Omega_{arag} < 1$, are highlighted with the colored background and the depth of $\Omega_{arag} = 1$ for the profile of the whole MPA is highlighted with a circle on the y-axis. Panel (**c**) shows the change between the 2090s and the 1990s in temperature (left; in °C) and oxygen (right; in mmol m⁻³). **d–o** Same as (**a**)–(**c**), but for **d–f** the proposed East Antarctic MPA, **g–i** the adopted Ross Sea region MPA, **j–l** the proposed Antarctic Peninsula MPA, **m–o** the adopted South Orkney Islands Southern Shelf MPA. Note that only the vertical profiles for the whole MPA are shown in panels (**m**)–(**o**) (see Methods). For vertical profiles for both decades of $\Omega_{calc}$ as well as temperature and oxygen, see Supplementary Figs. 1 and 6, respectively.

ranging from -0.01 in the Ross Sea region MPA to -0.07 in the East Antarctic MPA for the SSP5-8.5 scenario. This is caused by a pH decline that is up to five times stronger at the surface than at 1500 m across all MPAs for all except the lowest-emission scenario, for which the acidification at the surface and at depth are similar (solid lines in Fig. 4; same for $\Omega_{arag}$; see Supplementary Fig. 1 for $\Omega_{calc}$). Notably, this change in vertical gradient is less pronounced on the continental shelves in the Ross Sea region MPA and Weddell Sea MPA (dashed lines), reflecting the more efficient downward transfer of the OA signal where dense waters are formed and highlighting the key role of vertical mixing.

## Drastic twenty-first-century increase in calcium carbonate undersaturation

By 2100, the saturation state of aragonite and calcite decreases substantially in the Antarctic MPAs. While the depth of the aragonite saturation horizon, i.e., the depth at which $\Omega_{arag} = 1$, ranges from 351 to 678 m across the MPAs in the 1990s (deepest in the Weddell Sea), it is projected to rise to the surface in all MPAs by 2100 for the two highest-emission scenarios and to 40 to 70 m and 80 to 253 m for the lower-emission scenarios SSP2-4.5 and SSP1-2.6, respectively (see circles on the y-axis of all $\Omega_{arag}$ panels in Fig. 4). In the 1990s, 51–84% of waters in the upper 2000 m across all MPAs are undersaturated with respect to

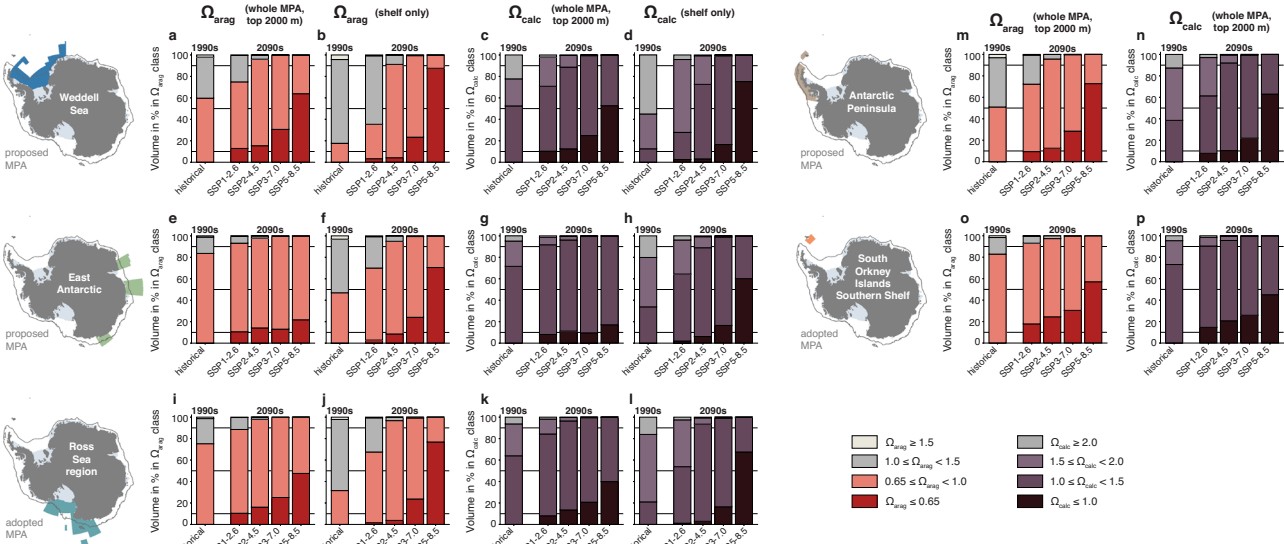

**Fig. 5 | Changes in the volume distribution across different $\Omega_{arag}$ and $\Omega_{calc}$ classes. a, b** Distribution of waters across different classes of the saturation state with respect to aragonite ($\Omega_{arag}$) **a** above 2000 m in the whole proposed Weddell Sea Marine Protected Area (MPA) and **b** on the continental shelves south of the 2000 m isobath (see dark gray contour in map). Results are shown for the 1990s of the historical simulation and for the 2090s in the four emission scenarios. **c, d** Same as (**a**) and (**b**), but for the distribution of waters across different classes of the saturation state with respect to calcite ($\Omega_{calc}$). Note the different classes of saturation states shown for aragonite and calcite. While the 0.65 threshold for $\Omega_{arag}$ was chosen to best highlight the stronger undersaturation on the shelf than in the whole MPA for the highest emission scenario, we assume that it is more costly for an organism to sustain calcification at $\Omega_{arag} < 0.65$ than at $0.65 \leq \Omega_{arag} < 1.0$ (ref. 122) and that this division of the volume of undersaturated waters is therefore meaningful. **e–l** Same as (**a**)–(**d**), but for **e–h** the proposed East Antarctic MPA and **i–l** the adopted Ross Sea region MPA. **m–p** Same as (**a**) and (**c**), but for **m** and **n** the proposed Antarctic Peninsula MPA and **o** and **p** the adopted South Orkney Islands Southern Shelf MPA. Note that only the distribution for the whole MPA is shown in (**m**)–(**p**) (see Methods).

aragonite ($\Omega_{arag} < 1$), whereas only 18–51% of continental shelf MPA waters experience undersaturation (Fig. 5). Furthermore, all waters in the MPAs are supersaturated with respect to calcite in the top 2000 m (see also Supplementary Fig. 2). However, by 2100, 17–63% of waters across all MPAs and 60–75% of all shelf waters are undersaturated with respect to calcite in the highest-emission scenario. More critically, virtually all (>95%) waters in the MPAs are undersaturated with respect to aragonite for the three highest-emission scenarios, with this ubiquitous undersaturation only being avoidable for the lowest-emission scenario SSP1-2.6 (see also Fig. 6). While 30–65% remain supersaturated for the shelves in this scenario, undersaturated conditions in the whole MPAs are widespread even under strong climate-change mitigation (>20% supersaturation only for the proposed Weddell Sea and Antarctic Peninsula MPAs but <10% for all others).

The onset of undersaturated conditions with respect to aragonite varies with depth and across seasons, scenarios and to some extent also regions (Figs. 6 and 7). Bottom waters averaged over the whole MPAs are undersaturated throughout the analysis period (on the continental shelf, only those in the Weddell Sea are initially supersaturated; second column in Fig. 6). In contrast, the saturation state in the top 200 m of the water column exceeds a value of 1 in all MPAs in the 1990s (ranging from 1.21 in the East Antarctic MPA to 1.39 in the Weddell Sea; solid lines in the first column of Fig. 6). By the 2090s, it decreases to 0.62–0.68 across regions for the highest-emission scenario and to 1.03–1.14 for the lowest-emission scenario. In all MPAs, the saturation state displays seasonal variability in the top ~50 m (dotted and dashed lines in Fig. 7a–e). As expected from the seasonal cycle of biological activity, sea surface temperature and the entrainment of carbon into the upper ocean, winter is generally projected to be undersaturated earlier than summer. For the highest-emission scenario SSP5-8.5, the onset of undersaturated conditions at the surface is fairly consistent across all MPAs (Fig. 7a–e), ranging from 2052 to 2060 for winter (June–August), from 2066 to 2069 for the annual mean and

from 2076 to 2085 for summer (December–February; latest in the Weddell Sea). Differences in the onset of undersaturation are larger across scenarios than across regions. For the emission scenario SSP3-7.0, surface waters become undersaturated in the annual mean between 2074 and 2083 across MPAs, with the proposed Antarctic Peninsula MPA and the South Orkney Islands Southern Shelf MPA being the only regions for which surface waters become undersaturated in the summer months before 2100. For the emission scenario SSP2-4.5, surface waters reach undersaturation within the twenty-first century during winter for the Ross Sea region MPA and the Antarctic Peninsula and East Antarctic MPAs. Under the lowest-emission scenario (SSP1-2.6), the upper ocean remains supersaturated with respect to aragonite across all MPAs until at least the end of the century.

Marine organisms are highly sensitive to the rate at which OA progresses, as this determines the time span available for possible acclimation or adaptation[78]. In the upper 50 m of the water column, our model projections suggest several decades of adaptation time between when 50% and 90% of all waters become undersaturated (Fig. 7f–j). For a given isoline of undersaturated volume, the number of years passing between the first and last month to reach this threshold varies across MPA regions and especially across emission scenarios (more compressed or elongated shape of isolines in Fig. 7f–j). While 20-28 years pass in the highest emission scenario between when the first (typically in late fall or winter) and last month (typically in late spring or summer) reach 50% undersaturation, this is 28–39 years across MPAs for the second-highest emission scenario SSP3-7.0. Given that for the SSP2-4.5 scenario, the threshold of 50% undersaturated volume is reached as early as 2060 for March and April in the Ross Sea region MPA, but not yet in summer by 2100, the time span between when the first and last month reach a given threshold appears to be longer the lower the emission scenario, providing ecosystems with an opportunity to acclimate or adapt to a changing environment under higher mitigation scenarios.

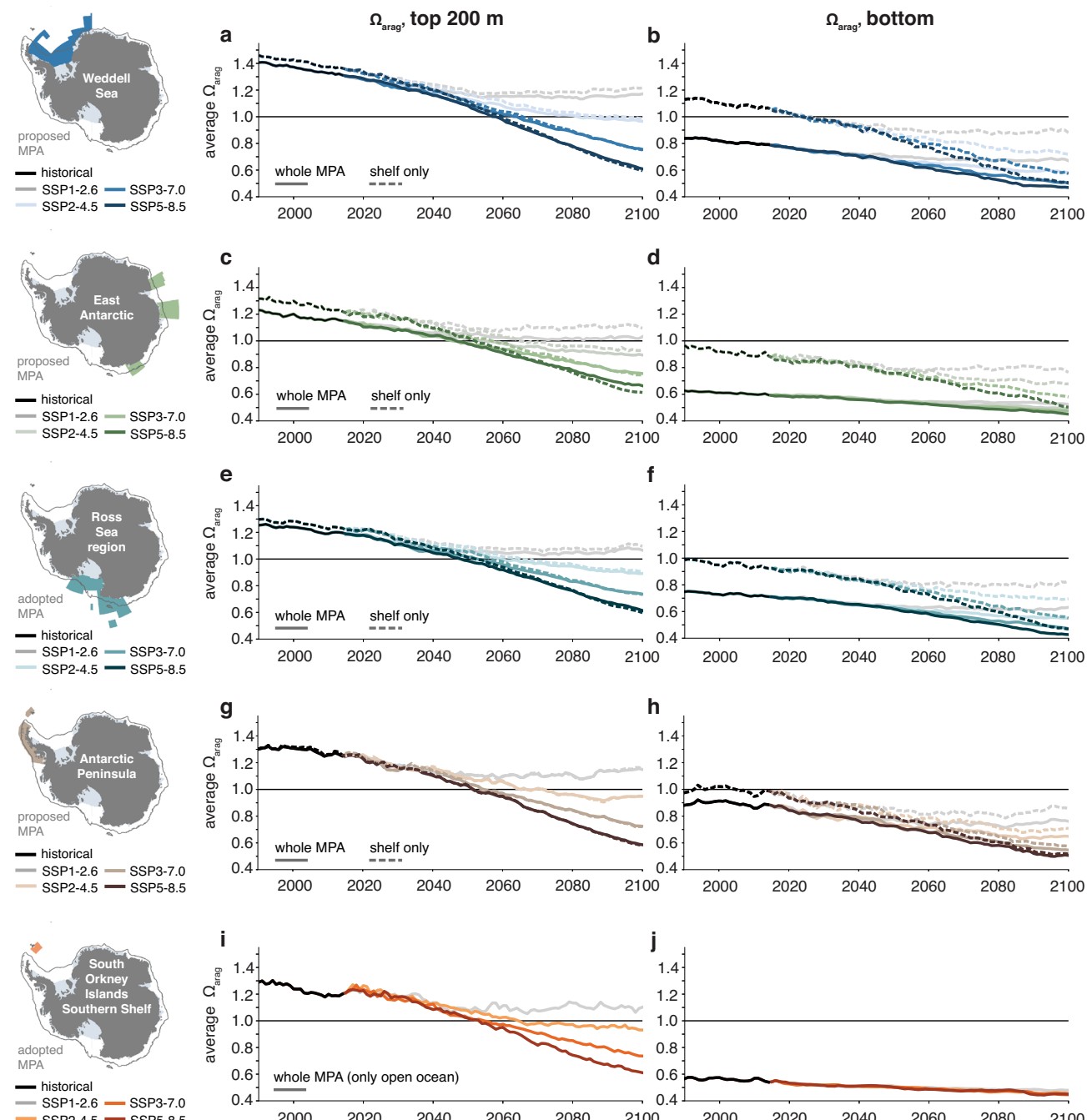

**Fig. 6 | Temporal evolution of $\Omega_{arag}$. a, b** Time series of **a** the top 200 m average saturation state with respect to aragonite ($\Omega_{arag}$) and **b** bottom $\Omega_{arag}$ in the proposed Weddell Sea Marine Protected Area (MPA) for the historical time period (black) and the SSP1-2.6 scenario (light gray), the SSP2-4.5 scenario (light blue), the SSP3-7.0 scenario (intermediate blue) and the SSP5-8.5 scenario (dark blue). Solid lines show the time series for the whole MPA and dashed lines for the continental shelves south of the 2000 m isobath (see dark gray contour in map). **c–j** Same as (**a**) and (**b**), but for **c** and **d** the proposed East Antarctic MPA, **e** and **f** the adopted Ross Sea region MPA, **g** and **h** the proposed Antarctic Peninsula MPA, **i** and **j** the adopted South Orkney Islands Southern Shelf MPA. Note that only the time series for the whole MPA are shown in (**i**) and (**j**) (see Methods). For the temporal evolution of $\Omega_{calc}$, see Supplementary Fig. 2.

## Climate-change feedbacks accelerate ocean acidification on the continental shelves

In addition to the evolution of atmospheric $CO_2$ levels, climate-change feedbacks, i.e., changes in the physical environment in response to atmospheric warming, will alter the projected rate of ocean acidification[2]. Computing the total climate effect from two simulations with varying and constant climate forcing, respectively (see Methods), the simulated net climate effect in the 2090s for the SSP5-8.5 emission scenario is to decrease pH on the continental shelf (red colors in

Fig. 8a, b). In the open ocean, the net climate effect decreases the top 200 m-average pH in most regions (Fig. 8a), but mostly increases bottom pH (Fig. 8b). The latter is likely due to a combination of less Antarctic Bottom Water formation, which results in less anthropogenic carbon reaching the abyss, and a southward shift of open-ocean water masses low in anthropogenic carbon following changes in atmospheric wind systems[73,74]. The only exceptions to this spatial pattern are downstream of the dense-water formation regions on the Weddell Sea and Ross Sea continental shelf (where climate change decreases

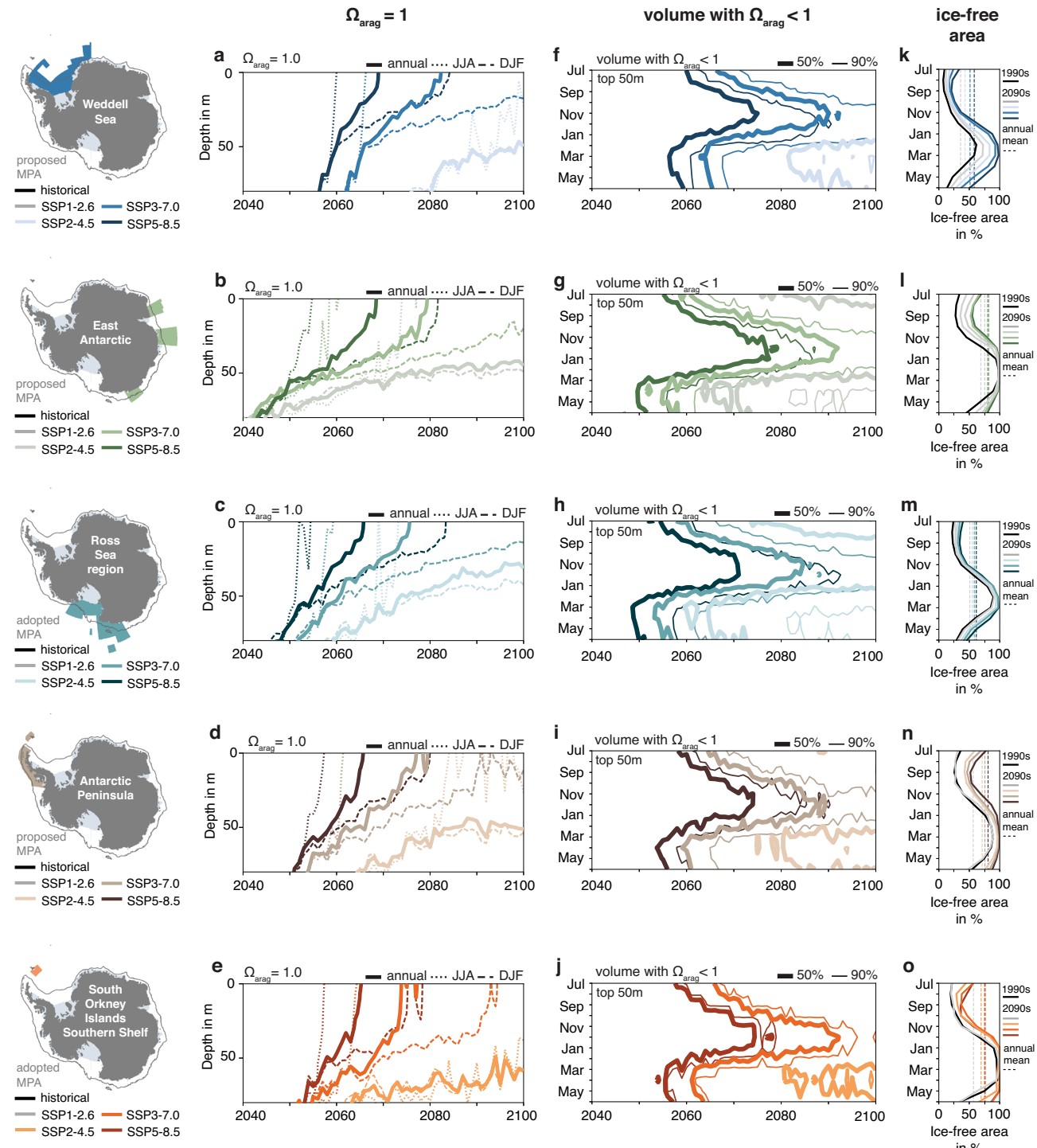

**Fig. 7 | Temporal evolution of $\Omega_{arag}$ and sea ice cover. a** Temporal evolution of the depth of the $\Omega_{arag} = 1$ isoline in the proposed Weddell Sea Marine Protected Area (MPA) for the SSP5-8.5 scenario (dark blue), the SSP3-7.0 scenario (intermediate blue) and the SSP2-4.5 scenario (light blue). The different line styles represent the annual mean (solid), the June–August mean (JJA; dotted) and the December–March mean (DJF; dashed). The SSP1-2.6 is not shown because the $\Omega_{arag} = 1$ isoline never reaches the top 80 m of the water column. **b–e** Same as (**a**), but for **b** the proposed East Antarctic MPA, **c** the adopted Ross Sea region MPA, **d** the proposed Antarctic Peninsula MPA and **e** the adopted South Orkney Islands Southern Shelf MPA. **f–j** Temporal evolution of the 50% (thick line) and 90% (thin line) isolines of the volume with $\Omega_{arag} < 1$ in the upper 50 m of the water column in the different MPAs. Colors show the different emission scenarios (see above). **k–o** Monthly ice-free surface area in the different MPAs for the 1990s (black) and the 2090s in the four emission scenarios (colors).

bottom pH; Fig. 8b) and some open-ocean regions in the Weddell Sea and East Antarctica for the top 200-m-average pH (where climate change increases top 200 m pH; Fig. 8a). In the latter regions, this can be attributed to enhanced upwelling due to increased winds, bringing waters to the upper ocean whose pH is higher than the pH of surface waters under this high-emission scenario[73,74]. Relative to the total change in pH between the 1990s and the 2090s, the climate-change effect is largest in the Weddell Sea and Ross Sea region MPAs, where it explains 19% and 16% (top 200 m average) and 20% and 19% (bottom), respectively, of the total change in pH on the continental shelf.

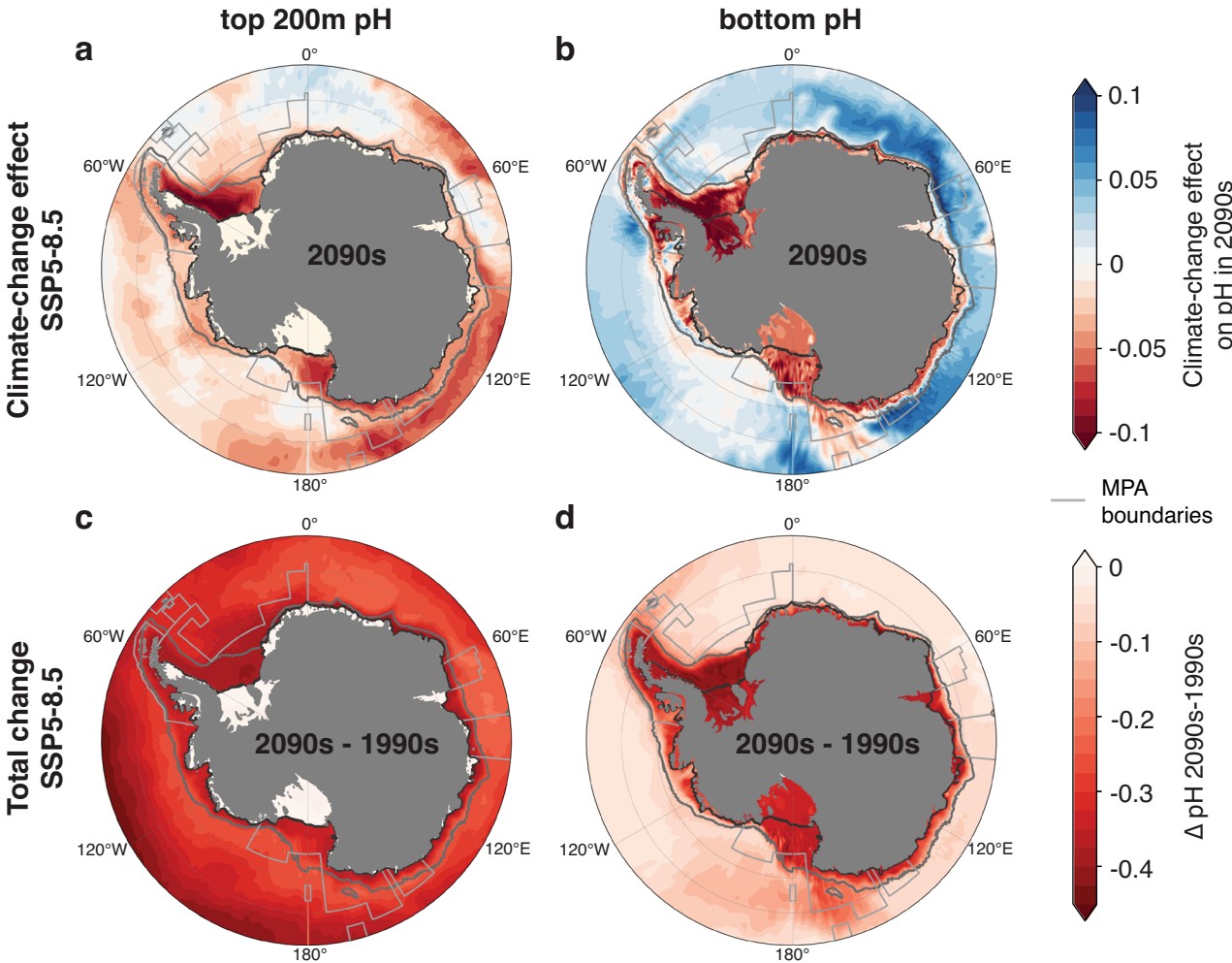

**Fig. 8 | Climate-change impacts on pH. a, b** Climate-change induced change in pH in the 2090s in the high-emission scenario SSP5-8.5 **a** averaged over the top 200 m of the water column and **b** at the seafloor. The contribution of climate change is calculated from the difference between the simulations *simA* and *simC*, i.e., the simulations with varying and constant climate, respectively (see Methods). **c, d** Total change in pH 2090s–1990s in the high-emission scenario SSP5-8.5 **c** averaged over the top 200 m of the water column and **d** at the seafloor. Outlines of the MPAs are shown in light gray, the gray contour denotes the 2000 m isobath, and the ice-shelf front is displayed in black.

## Discussion

Under the intermediate to high emission scenarios SSP2-4.5, SSP3-7.0 and SSP5-8.5, ecosystems in the shallow continental shelf seas of the proposed and adopted Antarctic MPAs will be exposed to severe OA by 2100. For the SSP3-7.0 and SSP5-8.5 scenarios, aragonite under-saturation is ubiquitous from surface to depth across all proposed and adopted MPAs by 2100, implying that aragonite-forming organisms such as pteropods[4,64,65,67] will be unable to find a refuge with super-saturated conditions and will hence be exposed to conditions that are chemically unstable for their shells. The Weddell Sea is sometimes referred to as a potential climate-change refuge, motivating the pro-posal to establish an MPA in this region[31]. Based on our model experiments, the volume of supersaturated water is indeed highest in this region in both the 1990s and the 2090s (Fig. 5), and OA is slightly delayed compared to the other MPAs over the twenty-first century (Fig. 7). However, by 2100, the severity of OA in the Weddell Sea is barely distinguishable from that in the other proposed and adopted MPAs for all except the lowest-emission scenario (Figs. 3–7). Strong local climate-change feedbacks explain why end-of-century OA in the Weddell Sea is comparable to that in other regions despite the delayed onset. While the area of the proposed Weddell Sea MPA has the lowest ice-free area of all MPAs in the 1990s (28% as compared to 48-61% in the annual mean, 57% as compared to 84–96% between December and

February), all regions are virtually ice free in summer by 2100 in the highest-emission scenario (Fig. 7k–o). In the Weddell Sea, the com-bined strong sea-ice retreat and an enhanced input of freshwater from ice-shelf basal melt[73,74] then facilitate accelerated oceanic $CO_2$ uptake according to the rise in atmospheric $CO_2$ (Fig. 8; see also Supple-mentary Figs. 3 and 4). Besides the reduced capping effect of sea ice for air-sea exchange following the sea-ice retreat, this can be attributed to the lower buffer capacity resulting in more oceanic $CO_2$ uptake per unit primary production in an acidified ocean[79] and, to a lesser extent, an overall increased primary production[73].

For other regions and the intermediate-emission scenario SSP2-4.5, the simulated climate-change feedback is smaller or even opposite in sign (Fig. 8 and Supplementary Figs. 3–5). While a smaller climate-change feedback can likely largely be attributed to a smaller decrease in sea-ice cover and a smaller increase in ice-shelf basal melt, respec-tively (Supplementary Fig. 4), the contribution of changes in $CO_2$ solubility and of a redistribution of water masses could also be a factor, especially where climate-change feedbacks act to increase pH (see, e.g., open-ocean regions near the Antarctic Peninsula in Supplemen-tary Fig. 5). Overall, this reflects the complex spatio-temporal interplay of the many factors contributing to the total climate-change feedback. Our model experiments were conducted with a more realistic process representation on the Antarctic continental shelf than earlier modeling

studies that investigated Southern Ocean climate-change feedbacks[2,80,81] and demonstrate that climate-change feedback on OA can be substantial under high-emission scenarios. While the projected upper ocean acidification under the high-emission scenario SSP5-8.5 in our model is very similar to the acidification previously reported for the high-latitude Southern Ocean in Earth system models[82], the bottom acidification rates on the Antarctic continental shelf are about two times larger in our model simulations (pH declines by more than 0.35 as opposed to by less than 0.2 in ref. [82]). Altogether, our findings illustrate the benefits of following a low-emission trajectory over the twenty-first century, which would minimize climate-change feedbacks on OA in the Southern Ocean.

Given the high biodiversity in MPAs[27,28], the projected severe OA under higher-emission scenarios will likely impact organisms on many trophic levels, ranging from primary producers (growth)[55] to fish (metabolic capacity and mortality)[3,70] and calcium carbonate-forming benthic organisms (shell dissolution; Fig. 1)[4,72]. Due to their immobility and often long lifespan[83], the impact on the latter will likely be substantial both on the continental shelves and in the open ocean, i.e., downstream of dense-water formation regions, where the OA signal spills from the shelf sea into the deep ocean (Fig. 2). While we have a limited understanding of adaptation and physiological acclimation in these organisms on multi-decadal time scales[78,84,85], it is conceivable that benthic organisms have a higher sensitivity to changing conditions than mobile pelagic organisms due to the smaller variability in pH and $\Omega$ at depth than in the surface layer (simulated monthly $\Omega_{arag}$ in the 1990s averaged over the Antarctic continental shelf spans 0.91-0.93 and 1.32-1.67 below 500 m and above 50 m, respectively). The exposure to smaller pH variability implies that benthic organisms likely have a smaller tolerance range, making them more susceptible to OA given their limited ability to move. We note that in the Weddell Sea, changes in the physical environment cause the transfer of newly formed dense waters to the abyss to be less efficient by 2100 under the highest-emission scenario[73,74], reducing the decadal rate of decline in bottom pH in the whole MPA from 0.03 (2040–2085) to 0.012 (2086–2100; based on a linear regression of the solid dark blue line in Fig. 3b). However, in the two highest emission scenarios, the projected pH by 2100 is <7.8 throughout the water column for all MPAs (Fig. 1), a level at which many detrimental effects have been reported for primary producers, zooplankton, fish and benthic organisms[3,4,67–71]. Even though it remains unclear in laboratory experiments to what extent a longer acclimation of adult organisms can reduce the severe impacts reported for egg development and juveniles[86], strong OA would almost certainly disrupt food webs, as OA might affect even those organisms that are not directly negatively impacted by OA themselves through direct OA effects on their prey or predators.

A disruption of Antarctic food webs seems even more likely when considering that OA is not the only marine ecosystem stressor exacerbated by anthropogenic climate change. High-latitude warming has been shown to aggravate OA impacts on Antarctic organisms[66,68,87], resulting in higher mortality and reduced hatching success of Antarctic dragonfish[68] and increased resting metabolic rates in Antarctic juvenile pteropods[66] than when considering OA alone. Furthermore, warming could push cold-adapted organisms in Antarctic coastal waters beyond their thermal tolerance by 2100[88] (for the projected temperature changes, see Fig. 4 and Supplementary Fig. 6). While our model projections indicate oxygen concentrations above the typically used hypoxic threshold of 63 mmol m$^{-3}$ [89] in the Antarctic MPAs for all scenarios (Supplementary Fig. 6), the projected decline in oxygen concentrations of up to ~50 mmol m$^{-3}$ might nonetheless negatively impact high-latitude ecosystems in conjunction with the projected warming of up to 1 °C and OA[90] (Fig. 4). The recent reduction in sea-ice cover due to the warming of waters around the Antarctic Peninsula has already negatively affected larval abundance of the Antarctic silverfish, which uses sea ice as spawning habitat[91], suggesting that the twenty-first-century sea-ice retreat projected for all proposed and adopted MPAs will negatively impact circumpolar abundances of this species (Fig. 7k–o and Supplementary Fig. 4).

Our twenty-first-century projections call for strong emission-mitigation efforts, as high-latitude Southern Ocean waters are only spared from severe OA in the lowest-emission scenario (SSP1-2.6) and as climate-change feedbacks substantially aggravate OA on the Antarctic continental shelf under the highest-emission scenario (SSP5-8.5). OA across all emission scenarios within the proposed and adopted MPAs is comparable to OA in continental shelf areas not protected within an MPA (Supplementary Fig. 7). This implies two things for the design and evaluation of MPAs: Firstly, as long as some portion of Antarctic waters remains outside of MPAs, these regions could serve as an important reference area for differentiating the ecosystem impacts of fishing from climate change and thus assessing the effectiveness of MPAs[29–31]. Secondly, given that ecosystems in high-latitude waters outside of the MPAs will be similarly affected by severe OA, a reduction of fishing activities in a larger fraction of Antarctic coastal waters could be considered to reduce cumulative stressors on the system and to preserve genetic, species and ecosystem diversity[47–49]. Future work should include a joint assessment of the evolution of all ecosystem stressors in Antarctic coastal waters over the twenty-first century to identify regions of greatest and lowest risk for marine organisms. In that context, our findings support the notion of today's Weddell Sea acting as a climate-change refuge[31], and the adoption of the Weddell Sea MPA should therefore have high priority. The active fisheries for Antarctic toothfish and Antarctic krill have been suggested to already be potentially impacting ecosystems in the Ross Sea[92,93] and near the Antarctic Peninsula[94,95], respectively. As climate change progresses, cumulative impacts from fishing and environmental change will thus become increasingly likely and possibly more devastating than the impacts of each stressor alone[96]. Therefore, better understanding the complex interactions of OA with other ecosystem stressors including fishing is key for sustainable management of Antarctic marine ecosystems under changing environmental conditions, which will likely make a larger fraction of the previously ice-covered high-latitude Southern Ocean accessible to fishing activities (Fig. 7). Altogether, the reduction of greenhouse gas emissions together with simultaneous management of fishing and other human activities, e.g., within MPAs, are necessary for safeguarding Antarctic marine ecosystems.

## Methods
### Description of FESOM-REcoM and all model experiments
For this study, we use the same model setup as refs. 73,74. We will summarize its main features hereafter, but refer the reader to the other publications for further detail. All model experiments are conducted with the global Finite Element Sea ice Ocean Model (FESOM) version 1.4[97], which includes a sea-ice[98] and an ice-shelf component[99] with constant ice-shelf geometry derived from RTopo-2[100]. The biogeochemical cycles of carbon, nitrogen, silicon, iron and oxygen are resolved by coupling FESOM to the Regulated Ecosystem Model version 2 (REcoM2)[101,102]. REcoM2 includes two phytoplankton groups (silicifying diatoms and a mixed small phytoplankton group) and two zooplankton groups and allows for variable stoichiometry.

The model equations are solved on a model grid with eddy-permitting resolution on Antarctic continental shelves (<5 km in southernmost parts; Supplementary Fig. 8). The grid resolution is lower in the open ocean (<150 km). 79% and 62% of all surface model grid cells are south of 60°S and 70°S, respectively. The model grid has 99 z-levels in the vertical. At the surface, all simulations are forced with 3-h atmospheric output from the model experiments of the AWI Climate Model (AWI-CM) that contributed to the "Coupled Model Intercomparison Project Phase 6 (CMIP6)"[76]. The model experiments are started in 1950 and initialized with output from the AWI-CM (physical tracers of FESOM) and with output from a simulation for the "Regional

Carbon Cycle Assessment and Processes 2 (RECCAP2)" project (biogeochemical tracers of REcoM2)[103].

For this study, eight model simulations are analyzed (4× *simA*, *simB*, 2× *simC*, *simD*) as outlined in the following: Following a historical simulation from 1950 to 2014, we simulate four emission scenarios for 2015–2100: SSP1-2.6 (lowest-emission scenario), SSP2-4.5, SSP3-7.0 and SSP5-8.5 (highest emission scenario). In each case, the ocean model receives fluxes of momentum, heat and freshwater from the 3-h atmospheric output corresponding to the given scenario and the atmospheric $CO_2$ boundary condition for the ocean model increases from 312 ppm in 1950[104] to 446, 603, 867 and 1135 ppm, respectively, by 2100[75] (*simA*). As such, *simA* represents the net carbon fluxes (the sum of the natural and anthropogenic components).

To isolate the anthropogenic carbon component, we conduct an additional simulation from 1950 to 2100 with varying climate, but with a constant atmospheric $CO_2$ concentration boundary condition from the year 1950 (*simD*). The anthropogenic component of the dissolved inorganic carbon (DIC) pool ($C_{anth}$) can then be computed as the difference between the DIC in *simA* and that in *simD*. Due to computational constraints, *simD* is only available for the high-emission scenario SSP5-8.5. For easier comparability with existing observation-based estimates, anthropogenic carbon concentrations are reported in µmol $kg^{-1}$ in the manuscript and we use monthly mean surface-referenced potential density to convert all model fields from the model units (mmol $m^{-3}$). We note that since *simD* was generated using the 1950 atmospheric $CO_2$ concentration, the anthropogenic carbon quantified in this study is only that which has accumulated since 1950 and not since pre-industrial times. Thus, the resulting model-based values of $C_{anth}$ are likely biased low compared to observation-based estimates which include all anthropogenic carbon. Yet, since most of the increase in atmospheric $CO_2$ levels occurred after 1950[104], our model-based $C_{anth}$ estimate can be assumed to capture the vast majority of the signal. The lower simulated bottom $C_{anth}$ concentrations in the East Antarctic open-ocean sector in the 1990s (<1 µmol $kg^{-1}$) than suggested by observations (up to 25 µmol $kg^{[-1,14,15,22,23]}$) may be attributed to the absence of shelf water masses, which are dense enough to efficiently transfer anthropogenic carbon to the abyss in this sector. However, we acknowledge the difficulty (a) to compare different observation-based estimates to each other due to the uncertainty arising from the use of various back-calculation techniques[18] and (b) to directly compare the observation-based estimates from various years to the decadal-average model tracer.

To assess the model drift, we run a control simulation *simB*, in which both climate variability and atmospheric $CO_2$ concentrations are held constant. For this experiment, the ocean model receives atmospheric $CO_2$ from the year 1950 and repeated year 1955 momentum, heat and freshwater fluxes. This year was identified to best represent "normal" atmospheric conditions, as determined from an assessment of the Southern Annular Mode and El Niño Southern Oscillation in the AWI-CM model output (see ref. 73 for details). Overall, the surface drift in carbonate chemistry variables is minimal, whereas some subsurface drift exists in *simB* (see Supplementary Fig. 9 for pH as an example). Any deep-ocean model drift can likely at least partially be attributed to a delayed adjustment of the deep ocean to upper ocean carbonate chemistry conditions. However, we note that since the model drift is generally much smaller than the acidification signal in the different emission scenarios, it only has a small impact on our results (with the notable exception of the lowest-emission scenario for the proposed Antarctic Peninsula MPA; Supplementary Fig. 9).

Furthermore, two complementary model experiments for 1950-2100 are performed in which climate variability is held constant, while atmospheric $CO_2$ concentrations increase (*simC*). By computing *simA* minus *simC* of any model tracer or flux, we can isolate the effect of climate change on the projected changes in these tracers or fluxes. Due to computational limitations, *simC* is only available for the

intermediate scenario SSP2-4.5 and the high-emission scenario SSP5-8.5. For this manuscript, we quantify the total climate effect, thus combining the impact of, e.g., warming and changes in circulation, sea-ice cover, freshwater input and biological productivity on the projected model fields (see Supplementary Figs. 3–5).

## Marine Protected Areas

Ocean acidification is assessed in five Marine Protected Areas (MPAs), of which two exist already (Ross Sea region and South Orkney Islands Southern Shelf) and three are proposed (Weddell Sea, East Antarctic, Antarctic Peninsula)[105]. Masks on the model grid for the different MPAs are derived from previously published coordinates[105,106]. For this study, we only consider the MPA area outside of the ice-shelf cavities and "continental shelf" and "open ocean" are separated at the 2000 m isobath. Based on the model grid, the surface area in each MPA corresponds to $1.44 \cdot 10^6$ $km^2$ for the Weddell Sea (53% on the continental shelf), $0.97 \cdot 10^6$ $km^2$ for the East Antarctic (29% on the continental shelf), $1.59 \cdot 10^6$ $km^2$ for the Ross Sea (45% on the continental shelf), $0.56 \cdot 10^6$ $km^2$ for the Antarctic Peninsula (73% on the continental shelf), $0.11 \cdot 10^6$ $km^2$ for the South Orkney Islands (0% on the continental shelf). As the South Orkney Islands Southern Shelf MPA is entirely defined as "open ocean" based on the model topography, we only report values for the whole MPA for this MPA in all figures. In total, 60% of the continental shelf area outside of the ice-shelf cavities is covered by the five MPAs assessed in this study. The area-weighted average grid resolution outside of the ice-shelf cavities corresponds to 35 km in the Weddell Sea MPA (spread 7–88 km; 7–55 km for the continental shelf; Supplementary Fig. 8), 46 km in the East Antarctic MPA (spread 6–93 km; 6–44 km for the continental shelf), 54 km in the Ross Sea region MPA (spread 4–138 km; 4–75 km for the continental shelf), 30 km in the Antarctic Peninsula MPA (spread 2–84 km; 2–45 km for the continental shelf) and 72 km in the South Orkney Islands Southern Shelf MPA (spread 57–85 km). We note that while there are between 1250 and 3632 model grid cells in the four largest MPAs, only 22 model grid cells lie within the South Orkney Islands Southern Shelf MPA.

## Calculation of carbonate chemistry

We calculate monthly mean pH and the saturation states of aragonite ($\Omega_{arag}$) and calcite ($\Omega_{calc}$) offline using the Python routines to model the ocean carbonate system (mocsy v2.0)[107] (code is available at https://github.com/jamesorr/mocsy, last access February 1, 2023). As input parameters to the functions, we use the monthly model output of dissolved inorganic carbon, alkalinity, potential temperature, salinity, silicic acid, phosphate (obtained from model nitrate field using the Redfield ratio 16) and sea-level pressure from the atmospheric output of the AWI-CM. The change in acidity reported in the manuscript is calculated from a change in hydrogen ion concentration ($[H^+]$) between the 2090s of a given emission scenario and the 1990s of the historical simulation.

## Data availability

Post-processed model output underlying the figures of this paper is deposited at https://doi.org/10.5281/zenodo.10291935[108]. The full model output underlying the findings of this study is deposited at the World Data Center for Climate (WDCC) at DKRZ under the following https://doi.org/10.26050/WDCC/FESOM14-REcoM2_A_hist_vA_vC (*simA*, historical)[109], https://doi.org/10.26050/WDCC/FESOM14-REcoM2_A_s126_vA_vC (*simA*, SSP1-2.6)[110], https://doi.org/10.26050/WDCC/FESOM14-REcoM2_A_s245_vA_vC (*simA*, SSP2-4.5)[111], https://doi.org/10.26050/WDCC/FESOM14-REcoM2_A_s370_vA_vC (*simA*, SSP3-7.0)[112], https://doi.org/10.26050/WDCC/FESOM14-REcoM2_A_s585_vA_vC (*simA*, SSP5-8.5)[113], https://doi.org/10.26050/WDCC/FESOM14-REcoM2_B_1921_cA_cC (*simB*)[114], https://doi.org/10.26050/WDCC/FESOM14-REcoM2_C_hist_vA_cC (*simC*, historical)[115], https://doi.org/10.26050/WDCC/FESOM14-REcoM2_C_s245_vA_cC (*simC*, SSP2-4.5)[116], https://doi.org/10.

26050/WDCC/FESOM14-REcoM2_C_s585_vA_cC (*simC*, SSP5-8.5)[117], https://doi.org/10.26050/WDCC/FESOM14-REcoM2_D_hist_cA_vC (*simD*, historical)[118], and https://doi.org/10.26050/WDCC/FESOM14-REcoM2_D_s585_cA_vC (*simD*, SSP5-8.5)[119].

## Code availability
The Fortran source code of FESOM1.4-REcoM2 can be obtained via https://fesom.de/models/fesom14/ (last accessed January 23, 2023). The code version used for the simulations analyzed in this study is deposited at https://doi.org/10.5281/zenodo.10290892[120]. The analysis of model output was done with the open-source software Python. The routines to compute the carbonate chemistry offline from the monthly model output in Python are available via https://github.com/jamesorr/mocsy (downloaded on February 24, 2022). All analysis scripts, including the version of the routines to compute the carbonate chemistry used in this study, can be accessed via https://doi.org/10.5281/zenodo.10295919[121].

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

## Acknowledgements

C.N. and M.H. acknowledge funding from the European Union's Horizon 2020 research and innovation programme under grant agreement No 820989 (project COMFORT). J.H. was supported by the Initiative and Networking Fund of the Helmholtz Association (Helmholtz Young Investigator Group Marine Carbon and Ecosystem Feedbacks in the Earth System [MarESys], grant number VH-NG-1301) and R.T. by the Helmholtz Climate Initiative REKLIM (Regional Climate Change), a joint research project of the Helmholtz Association of German research centres (HGF). C.N. and N.S.L. are grateful for funding from the U.S. Department of Energy (DE-SC0022243). C.M.B. acknowledges support from the Pew Charitable Trusts, National Science Foundation and NASA. Computing resources were provided by the North-German Super-computing Alliance (HLRN) project hbk00079. The work reflects only the authors' view; the European Commission and their executive agency are not responsible for any use that may be made of the information the work contains.

## Author contributions

C.N. and N.S.L. designed the study. C.N. ran the model simulations with support from R.T., M.H. and J.H. C.N. performed the analysis. C.M.B. provided the MPA masks. All authors contributed to the interpretation of the results. C.N. wrote the manuscript with input from all authors.

## Competing interests

The authors declare no competing interests.
