## [Peer Review File · Nature Communications]

Severe 21st-century ocean acidification in Antarctic Marine Protected AreasREVIEWER COMMENTS

Reviewer #1 (Remarks to the Author):

The study presents novel high-resolution ocean simulations investigating the impact of future climate change projections on ocean acidification of Antarctic Marine Protected Areas (MPAs). While the study has the potential to make a valuable contribution to ocean acidification (OA) and biodiversity in a future world, I recommend the study needs substantial revisions to realise its potential.

Following is a list of general issues I have with the paper and suggestions for improvement.

1. The focus on MPAs seems to be contrived, and the title and key conclusions are not clearly demonstrated in the study. For example, in the last sentence of the abstract - why does the study show the need "for continuity and expansion of MPAs" - the whole of Southern Ocean will experience similar OA. What makes expansion and continuity important? What does your study provide on the continuity question? I'm not convinced the title and the last sentence of the abstract have been clearly demonstrated.

2. The paper states the shelf regions experience faster OA than the open ocean. However, the simulations show much more nuanced behaviour. In the low emissions simulations, the open ocean has more OA than the shelf region. It is only the two high emissions where the shelves show more OA than the open ocean. I think this is an interesting result that warrants more investigation. Establishing MPAs may be important to preserving Antarctic biodiversity, but as you show, under high CO₂ emissions scenarios, the OA impacts will be severe. The more policy-relevant issue for Antarctic biodiversity is to limit our emissions. If biodiversity is on the shelf critical, then keeping to low emissions scenarios would substantially reduce the OA impacts and have the most benefit for the shelf regions. To be more relevant, I recommend focusing on how the different scenarios alter the trajectory and impact of OA in the open ocean and shelf regions. If we really go along the RCP8.5 scenario, the impacts will be catastrophic for many ecosystems, and the expansion of MPAs will not prevent this. The needed message is the benefit we gain by limiting our emissions.

Specific Comments

l3-4 there have been many studies of OA in the Southern Ocean. The unique bit here is the better resolution of the shelf regions. Use the improve resolution to provide new insights in the future OA.

l11-12 - need more analysis to show how this preserves ecosystem integrity

l18-19 - cite the Revelle factor

l33-34 - MPAs will comprise 60% of Antarctic Shelf water. Why does going to more areas improve resilience to OA?

l48-51 - What is the role of subduction of AABW on future OA impacts on these benthic ecosystems?

l68-69 - more severe OA on the shelf than the open ocean, not true for all emission scenarios

l81 - why did you lump the upper 2000 m of the ocean? biological reason? please justify.

l102-105 - to first order, OA is driven by atmospheric CO₂ and vertical ocean mixing

l114-115 - more severe OA on the shelf than the open ocean, not true for all emission scenarios

l121-122 - under the two high emission scenarios, aragonite is chemically unstable at the surface - this is a dramatic change in the chemical state of the ocean

l175-176 - dramatic OA in the high emission scenarios. Why is the OA in the Weddell much less than other regions in the low emissions scenarios? Preserving sea ice cover is important? is also important to preserving ice shelves and dramatic sea level rise.

l167-185 - the role of ocean warming and sea ice changes on the rate of OA and its impact.

l210 - I suggest either expanding the results to discuss and show dissolved oxygen simulations to remove this. Where did the 61.3 hypoxic threshold come from? how does these threshold change with emissions scenarios? is it avoided in the low emissions scenarios?

l217- Is there existing evidence of the impact of fishing on the Antarctic Ecosystems? What is it?

Figure 1. East Antarctica, Ross Sea and Weddell Sea MPAs show shelf areas have less OA impact than the open ocean in low emissions scenarios. Why? I think this is an important feature of the results that do require a high-resolution simulation to show. It also important if the goal is to preserve biodiversity in the shelf regions under OA.

Figure 2. Why upper 2000m?

Figure 3. similar to figure 1.

Reviewer #2 (Remarks to the Author):

Review of Severe 21st-century ocean acidification demands continuance and expansion of Antarctic

Marine Protected Areas by Nissen et al.

This paper uses a high-resolution ocean-sea ice-biogeochemistry model to explore ocean acidification in five Antarctic Marine Protected Areas (two adopted, three proposed) under four emission scenarios. Their major findings are that surface pH declines significantly in all MPAs by 2100, and to a greater extent from the lowest to the highest emissions scenario. They also show that by 2100 aragonite undersaturation occurs ubiquitously across all MPAs examined under all but the lowest emission scenario. They also investigate the effects of vertical mixing on ocean acidification and the transport of anthropogenic carbon to the subsurface and deep ocean, and projected changes in calcite undersaturation.

This work is of a very high quality and will make a substantial contribution to the field, due to its sound scientific approach and findings as well as its strong relevance to policy making. The work is timely in relation to decision making at international levels, both with respect to global climate change and protection of Antarctic marine systems. As well as being important in its own field, this paper will also be of interest and importance to a broader readership due to its broad relevance across the remit of global climate change and its impacts, and informing policy in this space. There are some minor issues and questions that I have outlined below, that I hope will help to improve the manuscript. As such, I believe that this work will be suitable for publication in Nature Communications, after minor/moderate revisions.

The methodology is robust in supporting the results reported and conclusions drawn, and is based on previously published work and ongoing world-leading research efforts. The data analysis, interpretation and conclusions reported in this paper are of a high standard in general, although I do have some points for clarification that I have outlined below. These require attention, but if satisfactorily addressed, they will certainly not prohibit publication. The methods description is sufficiently detailed for the purposes of this paper, and refers to other published work for more detail. One minor point here is that one of the references cited is Nissen et al., 2023, in review. If that paper is not published before this one, that reference will need to be removed.

In the following, I provide a small number of more major or overarching comments, followed by minor and editorial comments.

Major comments:

This work has the potential to be very valuable for scientific and broader audiences. However, I felt that there were opportunities missed to really emphasise this value and the key points of the work. In particular, the final paragraph of the main text (particularly lines 222-231) was not as compelling as it could have been. It feels a bit like a list of statements, rather than a well-crafted and coherent narrative. There are also a number of places where I felt that stronger wording could be used to really emphasise the importance of the work, for example “is advisable”, “are not spared”, “best policy tools”. I recommend that the authors revisit this section to really get the most impact out of their high-quality

scientific work.

Related to this, I felt that the framing of OA as one of multiple stressors on the system could be improved to be clearer and more compelling, both in the introduction (line 27-29; 59-62) and in the final paragraph (line 224-228). I suggest that more of a focus is placed on cumulative impacts (rather than just mentioning them) and the importance of increasing resilience in the system to the changes underway.

My final suggestion for a major improvement would be to explain more clearly what the different SSPs are and what they represent. I did not feel that this was adequately explained, in terms of the different emissions pathways and outcomes for CO₂ concentrations and warming, and what was explained was buried in the methods. Particularly if this work is to be used by people outside of the immediate field (e.g. other scientists, policy makers), I think it would really help in conveying the main messages of the paper if the SSPs were more clearly explained in the early stages of the main text.

Minor comments:

Title: I am not convinced continuance is right in this context. I suggest changing to continuation, which is more accurate.

Line 3: 'little is known'. Given what is known, I think this is probably an overstatement. I would suggest rewording to reflect that we do have some knowledge, but that significant gaps and uncertainties remain.

Line 7: here you make a compelling statement on changes in hydrogen ion concentration, but this is not addressed anywhere else in the main manuscript or supplementary information to substantiate this claim. This should be incorporated and at least mentioned in the main manuscript, or removed here.

Line 23-24: 'Antarctic shelf waters are a present-day sink for Canth'. I think it would be useful here to add a brief description of seasonal changes in sink/source dynamics, so that the reader has the full picture of what makes up the net annual sink.

Line 34: East Antarctic should be East Antarctica here (because of the preceding 'in').

Figure 1: I think it should be noted that the y-axis scale for the Weddell Sea is different to the others, and ideally, the ticks should include minimum and maximum bounds (rather than having any data points outside of the bounded scale).

Line 37: for the broad readership of Nature Communications, I think it is necessary to state what Phaeocystis is. Hence, I would add 'the haptophyte Phaeocystis' (or prymnesiophyte, although that may be too specific).

Line 53: should black be capitalised?

Line 57: there is considerable debate amongst the biological community as to whether there are any keystone species in the Southern Ocean (either Antarctic krill or silverfish), so I suggest the authors consider whether it is appropriate to use this terminology here without any substantiation. Especially the way the sentence is constructed 'keystone species, e.g. ... upper trophic level organisms' does not seem right to me without naming any organisms, so I suggest revising.

Line 61: among with. Should be either among or with, or possibly along with

Line 67-69: I suggest adding that continental shelf ecosystems are also more sensitive to the impacts of OA, to strengthen your point further.

Line 70-71: simulated top 2000 m average and bottom Canth concentrations. It becomes clear what this means later on, but on the first mention, it is ambiguous. I suggest spelling it out more clearly on first use and then using this terminology for the rest of the paper.

Figure 2: some of the text is too small to read; I suggest enlarging.

Line 91-92: severe shelf/MPA OA. I suggest spelling this out as 'severe OA on shelves and in the MPAs studied' or similar, to improve readability.

Line 99-100: ...up to five times stronger surface decline. This is awkward to read; I suggest rewording for clarity.

Line 121-123: 'it is projected to rise...'. Is this for all MPAs or a subset? It would help the reader's understanding to state this explicitly here.

Figure 5: text is too small for ice panels. Also, the label for 2100 appears to be misaligned in panel g

Line 149-151: I think it would be useful to state at the end of this sentence 'to at least end of century' or 'by 2100' or similar to really make it clear.

Line 153: adaptation. You could add acclimation here too.

Line 173-174: you could also point out here that this is also true for volume in the 2090s, as shown in figure 4, to further strengthen your case.

Line 182-185: would it also be worth stating here that the projected increase in PP also contributes to this effect?

Line 192: acclimatization. Is this correct or do you mean acclimation?

Line 191-196: after respectively (line 196), I suggest adding a statement that this means that benthos are likely to have a smaller tolerance range to variability in conditions and also mention that this is in addition to their limited ability to move.

Line 214: have should be has (referring to reduction).

Line 216: ...is worrisome. Related to my major comment about strengthening wording to be more compelling, I think it would be beneficial here to use stronger wording and to elaborate on what you mean.

Line 230: In addition to my major comment about this paragraph, I suggest adding 'and direct human impacts' after 'environmental change'.

Line 231: I suggest specifying 'global emissions of greenhouse gases'

Line 478: <100 km. From looking at the figure, this does not appear to be true for the region between 50 and 60 S in the Pacific and Atlantic sectors. Should it be >100 or <150?

Line 512: due should be due to

Line 531: state these what, so that this is not left hanging.

Line 542 (and 543, 544, 545): I suggest using standard scientific notation ($\times 10^6$) rather than Mio.

Lines 544-548: It is rather confusing that the authors state that 73% of the Antarctic Peninsula MPA area is on the shelf (i.e. 27% is not) and then to say that 'the vast majority' of grid points are defined as shelf. This also appears to go against what is stated in the text, which is that the whole of the Peninsula area is over the shelf. It strikes me that 27% is enough to run the analysis for the open ocean component.

However, I do trust the authors' judgement on this and their approach to the analysis, so I suspect it is just a case of this wording being confusing in the methods section. Please clarify.

References:

In general, there is very good coverage of the important references in this field and the study is placed appropriately within the wider literature. There are a small number of references that I felt were missing and would be beneficial to add for comprehensive coverage. These are:

Deppeler, S., Petrou, K., Schulz, K. G., Westwood, K., Pearce, I., Mckinlay, J., et al. (2018). Ocean acidification of a coastal Antarctic marine microbial community reveals a critical threshold for CO₂ tolerance in phytoplankton productivity. *Biogeosciences* 15, 209–231. doi: 10.5194/bg-15-209-2018

Gutt, J., Bertler, N., Bracegirdle, T. J., Buschmann, A., Comiso, J., Hosie, G., et al. (2015). The Southern ocean ecosystem under multiple climate change stresses – an integrated circumpolar assessment. *Glob. Change Biol.* 21, 1434–1453. doi: 10.1111/gcb.12794

Hancock, A. M., Davidson, A. T., Mckinlay, J., Mcminn, A., Schulz, K. G., and Van Den Enden, R. L. (2018). Ocean acidification changes the structure of an Antarctic coastal protistan community. *Biogeosciences* 15, 2393–2410. doi: 10.5194/bg-15-2393-2018

Henley, S. F., Cavan, E. L., Fawcett, S. E., Kerr, R., Monteiro, T., Sherrell, R. M., et al. (2020) Changing Biogeochemistry of the Southern Ocean and Its Ecosystem Implications. *Front. Mar. Sci.* 7:581. doi: 10.3389/fmars.2020.00581

Tortell, P. D., Payne, C. D., Li, Y. Y., Trimborn, S., Rost, B., Smith, W. O., et al. (2008). CO₂ sensitivity of Southern Ocean phytoplankton. *Geophys. Res. Lett.* 35:L04605.

Trimborn, S., Brenneis, T., Sweet, E., and Rost, B. (2013). Sensitivity of Antarctic phytoplankton species to ocean acidification: Growth, carbon acquisition, and species interaction. *Limnol. Oceanogr.* 58, 997–1007. doi: 10.4319/lo.2013.58.3. 0997

Westwood, K. J., Thomson, P. G., Van Den Enden, R. L., Maher, L. E., Wright, S. W., and Davidson, A. T. (2018). Ocean acidification impacts primary and bacterial production in Antarctic coastal waters during austral summer. *J. Exp. Mar. Biol. Ecol.* 498, 46–60. doi: 10.1016/j.jembe.2017.11.003

There are a number of places where the presentation of number ranges is grammatically incorrect. These should either be presented as 'between x and y', 'from x to y' or 'x-y'. Examples of incorrect constructions are at lines 179 (between December-February), 487 (from 1950-2014), 556 (between 1250-3632). Please correct these.

A minor grammatical point would be that there are lots of uses of an Oxford comma, which strictly speaking is incorrect and these should be removed. Examples are at lines 39 (after whales), 40 (after bivalves), 50 (after foraminifera), 490 (after heat), 534 (after input).

Supplementary information

Supp Figure 1: I suggest making the images larger to occupy some of the large amount of white space

and to make them easier to read.

Supp Figure 2: To improve readability, I suggest adding a y-axis scale to panel d to apply to panels d and e, in the same way that the scale on panel a applies to a, b and c. I would also suggest nudging the two lines of the y-axis label slightly further apart to avoid the 2s colliding. It would also help to make the labels 'annual mean' and 'DJF mean' more prominent for clarity.

Supp Figure 3: In line 3 of the caption, profile should be profiles.

Supp Figure 4: In line 5 of the caption, I think cb is a typo. I also think the y axis of panel b should say calc, not arag.

Supp Figure 5: in line 3 of the caption, the comma should be a semi-colon (;)

These comments and suggestions should all be amenable to being addressed by the authors, and in this case I believe the work would be ready for publication in Nature Communications. I am happy to review it again, or if the editor is happy handling this themselves, that is fine for me too.

Sian Henley

University of Edinburgh, UK

9/6/2023

Reviewer #3 (Remarks to the Author):

The study of Nissen and colleagues explores the implications provided by the high-resolution modelling of a much more severe OA in the Southern Ocean than reported ever before. The study excels in the modelling outputs, the scenarios are superbly linked to the physical-chemical process and produce state-of-the-art science on the ocean acidification over this century. The study really shows the importance of the model improvements and the implications it could bring.

On the other hand, the authors used the model outputs to suggest that the current predictions of the MPAs and its connectivity are not sufficient given the projected increases in OA. While this is a solid implication of the model results, the MPA efficiency cannot be evaluated only based on the saturation state (of 1) alone. As the MPAs are inherently linked to the protection of marine biology, the paper insufficiently covers the link between the chemistry and the biology, leaving a big gap in the interpretation. So far, in the literature, there are numerous studies for variety of species and functional groups for which specific pH or omega saturation state THRESHOLDS exists. Instead of just using a general $\omega = 1$ (which is even insufficient for the marine calcifiers impacted by dissolution), the authors have unique opportunity to apply the thresholds for different climate scenarios and show species sensitivity (or resilience). That would be a much stronger link to the MPAs due to climate change/OA and associated loss of habitat.

With that, I would suggest that the authors separate the pelagic and benthic habitats and assess the impacts separately, i.e. not only produced the change in the average over the upper 2000m, but actually more applicable for the pelagic and benthic biota, for example pteropods occupy the upper 200m, so project the threshold changes for that habitats, and equally the bottom for the adult benthic calcifiers. As it is now, it is not accurate enough in terms of biological representation. This should significantly

strengthen the implications of why the MPA expansion/reconsideration is needed.

The weakest point so far is the discussion on the MPA, implications, the only solution the enlargement along with the mitigation. There are several other options, like fisheries management, both in catch per unit effort change and timing of fisheries, etc. I think this section needs to be scientifically beefed-up content wise, best to examine the MPAs strategies and alternatives in the best managed MPAs across the world.

Response to comments by all reviewers

We thank all three reviewers for their time and thoughts spent on our paper. All comments helped to improve the paper. In summary, in the revised manuscript, we have made the following major changes:

- We have changed the title of the manuscript and reworked the discussion surrounding Marine Protected Areas. We think that the new pieces reflect the conclusions from our own analysis more clearly. We now highlight more clearly that both emission mitigation efforts and fisheries management, e.g., with MPAs, are needed to safeguard Antarctic ecosystems.
- In our writing, we put a larger emphasis on highlighting scenario differences, in particular the fact that the Antarctic continental shelf only experiences more severe ocean acidification than open ocean waters for the two highest-emission scenarios.
- We now include a more detailed description and discussion of climate-change feedbacks to illustrate their sizeable contribution to the severe ocean acidification on the Antarctic continental shelf under the highest-emission scenario.
- By reporting numbers for the top 200 m of the water column and the bottom layer in the revised manuscript (instead of for the top 2000m), we tie our findings more closely to habitats of calcifying organisms (pteropods in the top 200m and benthic organisms at the seafloor).
- We include new time-series figures of pH and the saturation states (Ω) with respect to aragonite and calcite in the manuscript. Thereby, we hope to make our findings more useful to scientists studying organisms who are sensitive to a threshold different from e.g., $\Omega=1$.

Reviewer #1 (Remarks to the Author):

The study presents novel high-resolution ocean simulations investigating the impact of future climate change projections on ocean acidification of Antarctic Marine Protected Areas (MPAs). While the study has the potential to make a valuable contribution to ocean acidification (OA) and biodiversity in a future world, I recommend the study needs substantial revisions to realise its potential.

We thank reviewer #1 for this feedback. In the revised version of the manuscript, we have implemented all suggestions, which improved the manuscript.

Following is a list of general issues I have with the paper and suggestions for improvement.

1. The focus on MPAs seems to be contrived, and the title and key conclusions are not clearly demonstrated in the study. For example, in the last sentence of the abstract - why does the study show the need "for continuity and expansion of MPAs" - the whole of Southern Ocean will experience similar OA. What makes expansion and continuity important? What does your study provide on the continuity question? I'm not convinced the title and the last sentence of the abstract have been clearly demonstrated.

One of the main challenges in the design, implementation and evaluation of Antarctic Marine Protected Areas (MPAs) is the scarcity of observational and modeling data on marine ecosystem stressors at high Southern Ocean latitudes. Since our model configuration generates a more realistic process representation on the Antarctic continental shelf than previous efforts,

we aim to make our model results directly usable for those designing and implementing Antarctic MPAs.

We agree that the manuscript would benefit from additional text describing the link between MPAs and environmental stressors such as ocean acidification (OA). MPAs promote genetic, species, and ecosystem diversity and ultimately resilience to other stressors such as ocean acidification through the prohibition or reduction of human activities such as fishing (e.g., Roberts et al., 2017). The projected severe OA by our model thus suggests continuation and expansion of Antarctic MPAs in order to increase ecosystem resilience to these environmental changes. If MPAs were stopped altogether and no other regulation of fishing activities was put in place, high-latitude ecosystems would thus be expected to be more vulnerable to climate change. Nonetheless, we agree with the reviewer on their second major comment (see below) that emissions reductions are equally important for minimizing impacts on marine ecosystems. Ultimately, both MPAs and emissions reductions are necessary to safeguard Antarctic marine ecosystems.

We realize that the link between our findings and the need for continuation and expansion of MPAs is of more indirect nature. To more adequately reflect this in the paper, we have changed the title of the manuscript to “Severe 21st-century ocean acidification in Antarctic Marine Protected Areas” and the second half of the abstract to now read:

“[...] By 2100, we project pH declines of up to 0.36 (total scale) for the top 200 m of the water column, corresponding to a 131% increase in hydrogen ion concentration relative to the 1990s. Vigorous vertical mixing of anthropogenic carbon on the continental shelves produces severe OA within the proposed Weddell Sea, Antarctic Peninsula and East Antarctic MPAs and the existing Ross Sea MPA, where pH decreases similarly throughout the water column. As a result, end-of-century aragonite undersaturation is ubiquitous across MPAs under the three highest emission scenarios. Given the cumulative threat to marine ecosystems by environmental change and direct human activities such as fishing, our findings call for strong emission-mitigation efforts and further management strategies to reduce pressures on ecosystem integrity, such as the continuation and expansion of Antarctic MPAs.”

Further, we have reworked the introduction to introduce the link more clearly between implementing MPAs and resilience to environmental change:

“By preserving genetic, species and ecosystem diversity through limitation or prohibition of human activities (e.g., fishing), the implementation of MPAs can increase resilience of marine ecosystems to environmental change, including those resulting from climate change⁴⁵.”

Lastly, we kindly refer the reviewer to our answer to the second major comment, which details how we put a more balanced emphasis on the need to reduce emissions of greenhouse gases and to continue and expand Antarctic MPAs.

2. The paper states the shelf regions experience faster OA than the open ocean. However, the simulations show much more nuanced behaviour. In the low emissions simulations, the open ocean has more OA than the shelf region. It is only the two high emissions where the shelves show more OA than the open ocean. I think this is an interesting result that warrants more investigation. Establishing MPAs may be important to preserving Antarctic biodiversity, but as you show, under high CO₂ emissions scenarios, the OA impacts will be severe. The more policy-relevant issue for Antarctic biodiversity is to limit our emissions. If biodiversity is on the shelf critical, then keeping to low emissions scenarios would substantially reduce the OA

impacts and have the most benefit for the shelf regions. To be more relevant, I recommend focusing on how the different scenarios alter the trajectory and impact of OA in the open ocean and shelf regions. If we really go along the RCP8.5 scenario, the impacts will be catastrophic for many ecosystems, and the expansion of MPAs will not prevent this. The needed message is the benefit we gain by limiting our emissions.

We thank the reviewer for this comment. As the reviewer correctly states, the different scenarios show vastly different OA trajectories for the 21st-century, which we did not emphasize enough in the original manuscript. Indeed, much can be gained (or much less may be lost) by reducing greenhouse gas emissions and committing to a low-emission trajectory for the remainder of the century.

In particular, the severity of the projected ocean acidification is much lower for the SSP1-2.6 and SSP2-4.5 scenarios than for the SSP3-7.0 and SSP5-8.5 scenarios, and as a result, waters on the Antarctic continental shelf are only more severely affected than open-ocean waters in the latter two scenarios. In addition to the different trajectories in atmospheric CO₂ concentration, the scenarios also differ in the strength of climate-change feedbacks, e.g., enhanced ocean acidification due to reduced sea-ice cover. In the highest-emission scenario, these feedbacks aggravate the projected ocean acidification by ~20%. In the revised manuscript, we now emphasize these scenario differences, in particular the avoidability of severe acidification under the lowest-emission scenario and the drivers of severe acidification under the highest-emission scenario.

The title in the first result section was changed to:

“More severe ocean acidification on continental shelves than in the open ocean under high-emission scenarios.”

Further, we have added statements to emphasize the scenario differences in the result and discussion sections:

“Over the 21st century, this shelf–open ocean gradient is sustained for the lower-emission scenarios, but it reverses for the two highest-emission scenarios in the regions encompassed by the Weddell Sea, East Antarctic and Ross Sea MPA [...].”

“For other regions and the intermediate-emission scenario SSP2-4.5, the simulated climate-change feedback is smaller or even opposite in sign (Fig. 8 and Supplementary Fig. 8). While a smaller climate-change feedback can likely largely be attributed to a smaller decrease in sea-ice cover and a smaller increase in ice-shelf basal melt, respectively (Supplementary Figures 4), the contribution of changes in CO₂ solubility and of a redistribution of water masses is possibly non-negligible, especially where climate-change feedbacks act to increase pH (see e.g., open-ocean regions near the Antarctic Peninsula in Supplementary Figures 5). Overall, this reflects the complex spatio-temporal interplay of the many factors contributing to the total climate-change feedback. Our model experiments, conducted with a more realistic process representation on the Antarctic continental shelf than earlier modeling studies that investigated Southern Ocean climate-change feedbacks^{2,76,77}, demonstrate that climate-change feedbacks on OA can be substantial under high-emission scenarios. While the projected upper ocean acidification under the high-emission scenario SSP5-8.5 in our model is very similar to the acidification previously reported for the high-latitude Southern Ocean in Earth system models⁷⁸, the bottom acidification rates on the Antarctic continental shelf are about two times larger in our model simulations (pH declines by more than 0.35 as opposed to by less than 0.2 in ref⁷⁸). Our

findings illustrate the benefits of following a low-emission trajectory over the 21st century, which would minimize climate-change feedbacks on OA.”

We also kindly refer the reviewer to our answer on the comment on L. 175-176, where we describe climate-change feedbacks in more detail and outline the changes applied to the result section.

Lastly, we fully agree with the reviewer that limiting our emissions is critical. Both emissions reductions and a reduction or prohibition of fishing activities will be key tools in the coming decades when aiming to reduce pressures on unique high-latitude marine ecosystems. As a number of uncertainties remain regarding the quantitative ecosystem-wide impacts of both ocean acidification and fishing (especially when these act in concert), our modeling results call for a) emissions reductions to minimize ocean acidification, b) more process studies to enhance our understanding of cumulative impacts of multiple ecosystem stressors and c) a simultaneous continuation of efforts to limit or prohibit fishing activities in high-latitude waters.

To clarify these points, we have reworked the final paragraph of the discussion in the revised manuscript, which now reads:

“Our 21st-century projections call for strong emission-mitigation efforts, as high-latitude Southern Ocean waters are only spared from severe OA in the lowest-emission scenario (SSP1-2.6) and as climate-change feedback substantially aggravate OA on the Antarctic continental shelf under the highest-emission scenarios (SSP5-8.5). OA across all emission scenarios within the proposed and adopted MPAs is comparable to OA in continental shelf areas not protected within an MPA (Supplementary Figure 7). This implies two things for the design and evaluation of MPAs: Firstly, as long as some portion of Antarctic waters remains outside of MPAs, these regions could serve as an important reference area for differentiating the ecosystem impacts of fishing from climate change and thus assessing the effectiveness of MPAs²⁷⁻²⁹. Secondly, given that ecosystems in high-latitude waters outside of the MPAs will be similarly affected by severe OA, a reduction of fishing activities in a larger fraction of Antarctic coastal waters could be considered to reduce cumulative stressors on the system. In that context, our findings support the notion of today’s Weddell Sea acting as a climate-change refuge²⁹, and the adoption of the Weddell Sea MPA should therefore have high priority. The active fisheries for Antarctic toothfish and Antarctic krill have been suggested to already be potentially impacting ecosystems in the Ross Sea^{88,89} and near the Antarctic Peninsula^{90,91}, respectively. As climate change progresses, cumulative impacts from fishing and environmental change will thus become increasingly likely and possibly more devastating than the impacts of each stressor alone⁹². Therefore, better understanding the complex interactions of OA with other ecosystem stressors including fishing is key for a sustainable management of Antarctic marine ecosystems under changing environmental conditions, which will likely make a larger fraction of the previously ice-covered high-latitude Southern Ocean accessible to fishing activities (Fig. 7). Altogether, the reduction of greenhouse gas emissions together with simultaneous management of fishing and other human activities, e.g., within MPAs, are necessary for safeguarding Antarctic marine ecosystems.”

Specific Comments

I3-4 there have been many studies of OA in the Southern Ocean. The unique bit here is the better resolution of the shelf regions. Use the improve resolution to provide new insights in the future OA.

We have rephrased the sentence as follows:

“Despite a particular sensitivity to ocean acidification (OA), significant gaps remain surrounding the future carbonate chemistry of waters on the Antarctic continental shelf.”

111-12 - need more analysis to show how this preserves ecosystem integrity

Please see our answer to the first major point above, where we explain how we have revised the wording in the abstract and elsewhere.

118-19 - cite the Revelle factor

Done as suggested. The sentence now reads:

“The Southern Ocean is especially sensitive to OA due to the prevalence of high CO₂ concentrations in cold waters, the upwelling of carbon-rich deep waters with low pH values and a high Revelle factor^{8,9,12}.”

133-34 - MPAs will comprise 60% of Antarctic Shelf water. Why does going to more areas improve resilience to OA?

This study assesses current (two) and proposed (three) MPAs. It is only with the proposed MPAs, which have not yet been adopted, that MPAs will comprise 60% of the Antarctic shelf water. As noted above, we made clearer in the text what role MPAs play in promoting resilience.

148-51 - What is the role of subduction of AABW on future OA impacts on these benthic ecosystems?

In Antarctic Bottom Water formation regions, OA signals are more quickly transferred to the bottom layer, both on the continental shelf and downstream in the open ocean. As a result, benthic ecosystems along these flow pathways will readily be exposed to OA. While the assumption of a higher susceptibility of these organisms to OA (relative to those in the pelagic realm) is conceivable due to their long lifespan and immobility, we have a limited understanding of adaptation and acclimation potential in these organisms.

We kindly refer the reviewer to the revised discussion section, in which we discuss the possibly high sensitivity of benthic organisms to ocean acidification, also in the context of AABW subduction:

“While we have a limited understanding of adaptation and physiological acclimation in these organisms on multi-decadal time scales^{74,77,78}, it is conceivable that benthic organisms have a higher sensitivity to changing conditions than of mobile pelagic organisms due to the smaller variability in pH and Ω at depth than in the surface layer (simulated monthly Ω_{arag} in the 1990s averaged over the Antarctic continental shelf spans 0.91-0.93 and 1.32-1.67 below 500 m and above 50 m, respectively). The exposure to smaller pH variability implies that benthic organisms likely have a smaller tolerance range, making them more susceptible to OA given their limited ability to move. We note that in the Weddell Sea, changes in the physical environment cause the transfer of newly formed dense waters to the abyss to be less efficient by 2100 under the highest-emission scenario^{69,70}, reducing the decadal rate of decline in bottom pH in the whole MPA from 0.03 (2040-2085) to 0.012 (2086-2100; based on a linear regression of the solid dark blue line in Fig. 3b). However, in the two highest emission scenarios, the projected pH by 2100 is <7.8 for all MPAs (Fig. 1), a level at which many detrimental effects have been reported for primary producers, zooplankton, fish and benthic organisms^{3,4,63-67}.”

168-69 - more severe OA on the shelf than the open ocean, not true for all emission scenarios
 Good catch! In the revised version of the manuscript, we provide more detail on scenario differences and highlight these differences more clearly (including climate-change feedbacks for the SSP2-4.5 and SSP5-8.5 scenarios). We refer the reviewer to our answer to the second major point above, which details all the modifications done in the manuscript in response to this point.

181 - why did you lump the upper 2000 m of the ocean? biological reason? please justify.
 This choice was made to more easily synthesize our findings.

We agree with the reviewer that the link between changes in water chemistry and biology at different depths could be made clearer for the reader. In the revised manuscript, we report our findings for the top 200m of the water column and for the bottom layer to better reflect benthic and pelagic habitats of calcifying organisms such as pteropods and benthic ecosystems.

We have further modified Figure 1 and Figure 2 of the manuscript (see Figs. 1 & 2 below). All numbers reported in the text were adapted accordingly. Please note that in Figure 2, we have further decided to also show panels for the regions “all shelves” and “all open ocean” for a circumpolar view on the invasion of anthropogenic carbon into the ocean.

Fig. 1: Revised Fig. 1 of the manuscript.

Fig. 2: Revised Figure 2 of the manuscript.

I102-105 - to first order, OA is driven by atmospheric CO₂ and vertical ocean mixing
 We agree with the reviewer and have adapted the respective sentence as follows:

“Notably, this change in vertical gradient is less pronounced on the continental shelves in the Ross Sea MPA and Weddell Sea MPA (dashed lines), reflecting the more efficient downward transfer of the OA signal where dense waters are formed and highlighting the key role of vertical mixing.”

I114-115 - more severe OA on the shelf than the open ocean, not true for all emission scenarios
 We have thoroughly reworked this aspect of the manuscript, and we refer the reviewer to our answer to the second major point above for all changes.

I121-122 - under the two high emission scenarios, aragonite is chemically unstable at the surface - this is a dramatic change in the chemical state of the ocean

We entirely agree with the reviewer on this point. To keep the focus of this part of the manuscript on the results, we have not made any change to the indicated lines. However, we would like to point the reviewer to the revised discussion section, in which we now state:

“Aragonite undersaturation is ubiquitous from surface to depth across all proposed and adopted MPAs by 2100, implying that aragonite-forming organisms such as pteropods^{11,59,60,62} will be unable to find a refuge with supersaturated conditions and will hence be exposed to conditions that are chemically unstable for their shells.”

1175-176 - dramatic OA in the high emission scenarios. Why is the OA in the Weddell much less than other regions in the low emissions scenarios? Preserving sea ice cover is important? is also important to preserving ice shelves and dramatic sea level rise.

We thank the reviewer for this comment. Indeed, our modeling results highlight that preserving sea-ice cover and keeping ice-shelf basal melt rates low are important factors to minimizing ocean acidification. In general, a reduced sea-ice cover exposes a larger ocean surface area to the atmosphere, facilitating additional oceanic uptake of anthropogenic CO₂ and hence aggravating ocean acidification. Similarly, a higher input of freshwater from ice-shelf basal melting reduces the buffer capacity of near-coastal waters and increases ocean acidification. In the present-day, the Weddell Sea has the highest sea-ice cover of all five MPA regions (in agreement with observations, see e.g., Eayrs et al., 2019). When this sea-ice cover is significantly reduced towards the end of the 21st-century in the highest-emission scenarios, a much larger surface area in the Weddell Sea is exposed to much higher atmospheric CO₂ levels, resulting in accelerated ocean acidification rates for this region. This explains why by the year 2100, the Weddell Sea does not act as the climate-change refuge anymore that it is today (see e.g., Teschke et al., 2021 and our manuscript).

To put more emphasis on the dynamics at play and to illustrate ocean acidification feedbacks involving the physical climate system, we have dedicated a new sub-section in the result section to this topic. In particular, we have added new figures to the manuscript (see Figures 3 & 4 below, corresponding to Figure 8 and Supplementary Figure 5 of the revised manuscript) and added the following text:

“Climate-change feedbacks accelerate ocean acidification on the continental shelves. In addition to the evolution of atmospheric CO₂ levels, climate-change feedbacks, i.e., changes in the physical environment in response to the atmospheric warming, will alter the projected rate of ocean acidification². Computing the total climate effect from two simulations with varying and constant climate forcing, respectively (see Methods), the simulated net climate effect in the 2090s for the SSP5-8.5 emission scenario is to decrease pH on the continental shelf (red colors in Fig. 8a & b). In the open ocean, the net climate effect decreases the top 200 m-average pH in most regions (Fig. 8a), but mostly increases bottom pH (Fig. 8b). The latter is likely due to a combination of less Antarctic Bottom Water formation, which results in less anthropogenic carbon reaching the abyss, and a southward shift of open-ocean water masses low in anthropogenic carbon following changes in atmospheric wind systems^{69,70}. The only exceptions to this spatial pattern are downstream of the dense-water formation regions on the Weddell Sea and Ross Sea continental shelf (where climate change decreases bottom pH; Fig. 8b) and some open-ocean regions in the Weddell Sea and East Antarctica for the top 200 m-average pH (where climate change increases top 200 m pH; Fig. 8a). In the latter regions, this can be attributed to enhanced upwelling due to increased winds, bringing waters to the upper ocean whose pH is higher than the pH of surface waters under this high-emission scenario^{69,70}. Relative to the total change in pH between the 1990s and the 2090s, the climate-change effect is largest in the Weddell Sea and Ross Sea MPAs, where it explains 19% and 16% (top 200 m average) and 20% and 19% (bottom), respectively, of the total change in pH on the continental shelf.”

In the discussion section, we have further elaborated on this point, and the text reads:

“Besides the reduced capping effect of sea ice for air-sea exchange following the sea-ice retreat, this can be attributed to the lower buffer capacity resulting in more oceanic CO₂ uptake per unit primary production in an acidified ocean⁷⁵ and, to a lesser extent, an overall increased

primary production⁶⁹. For other regions and the intermediate-emission scenario SSP2-4.5, the simulated climate-change feedback is smaller or even opposite in sign (Fig. 8 and Supplementary Fig. 8). While a smaller climate-change feedback can likely largely be attributed to a smaller decrease in sea-ice cover and a smaller increase in ice-shelf basal melt, respectively (Supplementary Figures 4), the contribution of changes in CO₂ solubility and of a redistribution of water masses is possibly non-negligible, especially where climate-change feedbacks act to increase pH (see e.g., open-ocean regions near the Antarctic Peninsula in Supplementary Figures 5). Overall, this reflects the complex spatio-temporal interplay of the many factors contributing to the total climate-change feedback. Our model experiments, conducted with a more realistic process representation on the Antarctic continental shelf than earlier modeling studies that investigated Southern Ocean climate-change feedbacks^{2,76,77}, demonstrate that climate-change feedbacks on OA can be substantial under high-emission scenarios. While the projected upper ocean acidification under the high-emission scenario SSP5-8.5 in our model is very similar to the acidification previously reported for the high-latitude Southern Ocean in Earth system models⁷⁸, the bottom acidification rates on the Antarctic continental shelf are about two times larger in our model simulations (pH declines by more than 0.35 as opposed to by less than 0.2 in ref⁷⁸). Our findings illustrate the benefits of following a low-emission trajectory over the 21st century, which would minimize climate-change feedbacks on OA.”

Fig. 3: New Fig. 8 in the revised manuscript.

Fig. 4: New Supplementary Figure 5 in the revised manuscript.

I167-185 - the role of ocean warming and sea ice changes on the rate of OA and its impact. Please see our comment above for a detailed answer on this point.

I210 - I suggest either expanding the results to discuss and show dissolved oxygen simulations to remove this. Where did the 61.3 hypoxic threshold come from? how does these threshold change with emissions scenarios? is it avoided in the low emissions scenarios?

The threshold $61.3 \text{ mmol kg}^{-1}$ (or 63 mmol m^{-3}) cited in the discussion section is the threshold commonly used to delineate well-oxygenated waters from hypoxic waters in the literature (e.g., Breitburg et al., 2018). If oxygen concentrations fall below this threshold, negative impacts on marine organisms are often considered to become increasingly likely (Breitburg et al., 2018). We acknowledge that critical oxygen thresholds are highly species-specific and any species' ability to acclimatize and adapt to changing oxygen conditions will also be impacted by the rate of change and the variability in oxygen conditions it is exposed to. Even though the main topic of our paper is ocean acidification, this environmental stressor does not act in isolation in the future ocean, and it is important to discuss warming and deoxygenation together with acidification. To better highlight the importance of these cumulative impacts (also in response to a comment by reviewer #2, see below), we have included the depth profiles of temperature

change and oxygen change in the revised manuscript (Fig. 5 below, corresponding to Fig. 4 in the revised manuscript).

In the discussion section, we have slightly revised the sentence in question:

“While our model projections indicate oxygen concentrations above the typically used hypoxic threshold of 63 mmol m^{-3} ⁸² in the Antarctic MPAs for all scenarios (Supplementary Fig. 3), the projected decline in oxygen concentrations of up to $\sim 50 \text{ mmol m}^{-3}$ might nonetheless negatively impact high-latitude ecosystems in conjunction with the projected warming of up to 1°C and OA^{83} (see last column in Fig 4).”

Fig. 5: New Figure 4 in the revised manuscript.

I217- Is there existing evidence of the impact of fishing on the Antarctic Ecosystems? What is it? There are primarily two active fisheries in the high-latitude Southern Ocean: Antarctic krill and Antarctic toothfish (e.g., Ainley & Pauly 2013; Ryan et al., 2023). While uncertainty surrounding

the number, size, connectivity, and ecosystem dynamics of toothfish populations throughout the Southern Ocean has to be acknowledged, previous studies for the Ross Sea have suggested that fishing for toothfish has impacted Weddell Seals, Ross Sea killer whales, Adelie penguins, and potentially other species (e.g., Ainley et al., 2017; Salas et al., 2017). Similarly, fishing for Antarctic krill, which occurs primarily adjacent to the western Antarctic Peninsula and Scotia Sea, has grown more concentrated in coastal areas, increasingly overlapping with predators like penguins and whales (Trivelpiece et al., 2011; Ainley & Pauly 2013; Hinke et al., 2017; Ryan et al., 2023). According to recent research, the cumulative catch of Antarctic krill may be greater than the amount consumed by local predators, and greater than the local replenishable population of krill (Trathan et al., 2022). Given that krill is an important grazer for high-latitude phytoplankton (e.g., Steinberg & Landry, 2017), such fishing-induced changes in krill abundances can therefore be expected to also impact the base of the foodweb. Under climate change, a larger fraction of the high-latitude Southern Ocean is projected to be ice free (see our results), making these regions more accessible to fishing vessels. Therefore, to limit future pressures on high-latitude ecosystems, both emission mitigation and fisheries management (e.g., via a network of MPAs) are necessary.

We have added a sentence on the known impacts of fishing to the revised discussion section:

“The active fisheries for Antarctic toothfish and Antarctic krill have been suggested to already be potentially impacting ecosystems in the Ross Sea^{87,88} and near the Antarctic Peninsula^{89,90}, respectively.”

Figure 1. East Antarctica, Ross Sea and Weddell Sea MPAs show shelf areas have less OA impact than the open ocean in low emissions scenarios. Why? I think this is an important feature of the results that do require a high-resolution simulation to show. It also important if the goal is to preserve biodiversity in the shelf regions under OA.

In the present-day and under lower-emission scenarios SSP1-2.6 and SSP2-4.5, the simulated average pH is higher on the continental shelves than in the open ocean both when averaging over the top 2000 m of the water column (as done in the original submission) and when averaging over the top 200 m (as in the revised version, see Fig. 1 above, see also our answer to the reviewer’s comment on L. 81). We note that the difference in pH between the continental shelves and the open ocean is much smaller when only averaging over the top 200 m. This can be understood with bottom topography and the vertical profile of pH (in general, higher at the surface than at greater depths, see Fig. 4 of the revised paper). The continental shelves are typically much shallower than 2000m, such that when averaging over the top 2000m, a much higher volume of water (relatively speaking) is from the top 1000m for the continental shelves than for open ocean, skewing the comparison between these two regions towards higher pH on the continental shelf than in the open ocean. We attribute the differences that remain when averaging over the top 200m to differences in sea-ice cover, which shield the high-latitude coastal regions from anthropogenic CO₂ uptake relative to regions in the open ocean. We realize that the way of presenting our results in the submitted manuscript might have been misleading. Therefore, and in response to a comment by reviewer #3 (on the difficulty to relate changes in water chemistry to biological impacts when averaging over a large body of water such as the top 2000m, see below), we report averages separately for the top 200m and the bottom in the revised manuscript.

Please see our answer to the reviewer’s comment on L. 81 for changes in the text. Further, we refer to the reader to our answer to the reviewer’s comment on L. 175-176, in which we detail how we have included the impact of climate change on ocean acidification, such as via changes in sea-ice cover and ice-shelf basal melt rates.

Figure 2. Why upper 2000m?

We kindly refer the reviewer to our answer on the comment on L. 81 above.

Figure 3. similar to figure 1.

Please see our answer to the reviewer's comment on L. 175-176.

Reviewer #2 (Remarks to the Author):

Review of Severe 21st-century ocean acidification demands continuance and expansion of Antarctic Marine Protected Areas by Nissen et al.

This paper uses a high-resolution ocean-sea ice-biogeochemistry model to explore ocean acidification in five Antarctic Marine Protected Areas (two adopted, three proposed) under four emission scenarios. Their major findings are that surface pH declines significantly in all MPAs by 2100, and to a greater extent from the lowest to the highest emissions scenario. They also show that by 2100 aragonite undersaturation occurs ubiquitously across all MPAs examined under all but the lowest emission scenario. They also investigate the effects of vertical mixing on ocean acidification and the transport of anthropogenic carbon to the subsurface and deep ocean, and projected changes in calcite undersaturation.

This work is of a very high quality and will make a substantial contribution to the field, due to its sound scientific approach and findings as well as its strong relevance to policy making. The work is timely in relation to decision making at international levels, both with respect to global climate change and protection of Antarctic marine systems. As well as being important in its own field, this paper will also be of interest and importance to a broader readership due to its broad relevance across the remit of global climate change and its impacts, and informing policy in this space. There are some minor issues and questions that I have outlined below, that I hope will help to improve the manuscript. As such, I believe that this work will be suitable for publication in Nature Communications, after minor/moderate revisions.

The methodology is robust in supporting the results reported and conclusions drawn, and is based on previously published work and ongoing world-leading research efforts. The data analysis, interpretation and conclusions reported in this paper are of a high standard in general, although I do have some points for clarification that I have outlined below. These require attention, but if satisfactorily addressed, they will certainly not prohibit publication. The methods description is sufficiently detailed for the purposes of this paper, and refers to other published work for more detail. One minor point here is that one of the references cited is Nissen et al., 2023, in review. If that paper is not published before this one, that reference will need to be removed.

In the following, I provide a small number of more major or overarching comments, followed by minor and editorial comments.

We thank Sian Henley for the positive feedback and for the detailed comments, which contributed to improving the paper. Please see our answers to the comments below.

We note that in the meantime, the paper "Nissen et al., 2023, in review" has been published in *Journal of Climate*:

Nissen, C., Timmermann, R., Hoppema, M., & Hauck, J. (2023). A Regime Shift on Weddell Sea Continental Shelves with Local and Remote Physical and Biogeochemical Implications is Avoidable in a 2°C Scenario. *Journal of Climate*, 36(19), 6613–6630. <https://doi.org/10.1175/JCLI-D-22-0926.1>

The reference in the revised manuscript was updated accordingly.

Major comments:

This work has the potential to be very valuable for scientific and broader audiences. However, I felt that there were opportunities missed to really emphasise this value and the key points of the work. In particular, the final paragraph of the main text (particularly lines 222-231) was not as compelling as it could have been. It feels a bit like a list of statements, rather than a well-crafted and coherent narrative. There are also a number of places where I felt that stronger wording could be used to really emphasise the importance of the work, for example “is advisable”, “are not spared”, “best policy tools”. I recommend that the authors revisit this section to really get the most impact out of their high-quality scientific work.

We have revised the paragraph in question to strengthen the wording, and it now reads:

“Our 21st-century projections call for strong emission-mitigation efforts, as high-latitude Southern Ocean waters are only spared from severe OA in the lowest-emission scenario (SSP1-2.6) and as climate-change feedback substantially aggravate OA on the Antarctic continental shelf under the highest-emission scenarios (SSP5-8.5). OA across all emission scenarios within the proposed and adopted MPAs is comparable to OA in continental shelf areas not protected within an MPA (Supplementary Figure 7). This implies two things for the design and evaluation of MPAs: Firstly, as long as some portion of Antarctic waters remains outside of MPAs, these regions could serve as an important reference area for differentiating the ecosystem impacts of fishing from climate change and thus assessing the effectiveness of MPAs²⁷⁻²⁹. Secondly, given that ecosystems in high-latitude waters outside of the MPAs will be similarly affected by severe OA, a reduction of fishing activities in a larger fraction of Antarctic coastal waters could be considered to reduce cumulative stressors on the system. In that context, our findings support the notion of today’s Weddell Sea acting as a climate-change refuge²⁹, and the adoption of the Weddell Sea MPA should therefore have high priority. The active fisheries for Antarctic toothfish and Antarctic krill have been suggested to already be potentially impacting ecosystems in the Ross Sea^{88,89} and near the Antarctic Peninsula^{90,91}, respectively. As climate change progresses, cumulative impacts from fishing and environmental change will thus become increasingly likely and possibly more devastating than the impacts of each stressor alone⁹². Therefore, better understanding the complex interactions of OA with other ecosystem stressors including fishing is key for a sustainable management of Antarctic marine ecosystems under changing environmental conditions, which will likely make a larger fraction of the previously ice-covered high-latitude Southern Ocean accessible to fishing activities (Fig. 7). Altogether, the reduction of greenhouse gas emissions together with simultaneous management of fishing and other human activities, e.g., within MPAs, are necessary for safeguarding Antarctic marine ecosystems.”

Further, we have rephrased the sentence on L. 216 of the submitted manuscript (“is worrisome”) to now read:

“The recent reduction in sea-ice cover due to the warming of waters around the Antarctic Peninsula has already negatively affected larval abundance of the Antarctic silverfish, which

uses sea ice as spawning habitat⁸⁶, suggesting that the 21st-century sea-ice retreat projected for all MPAs will negatively impact circumpolar abundances of this species (Fig. 7k-o and Supplementary Fig. 8)."

Related to this, I felt that the framing of OA as one of multiple stressors on the system could be improved to be clearer and more compelling, both in the introduction (line 27-29; 59-62) and in the final paragraph (line 224-228). I suggest that more of a focus is placed on cumulative impacts (rather than just mentioning them) and the importance of increasing resilience in the system to the changes underway.

We appreciate the reviewer's feedback. In response to this comment, we have made several changes to our manuscript. In particular, we have modified a part of the introduction, which now reads:

"By preserving genetic, species and ecosystem diversity through limitation or prohibition of human activities (e.g., fishing), the implementation of MPAs can increase resilience of marine ecosystems to environmental change, including those resulting from climate change⁴⁵. OA is one of many Southern Ocean ecosystem stressors arising from climate change, along with warming, deoxygenation and changes in sea-ice cover and circulation. As such, a change in each of these factors individually or in a multitude of factors combined directly impacts the fitness of marine organisms and ecosystem functioning^{27,46-48}. A large number of studies [...]"

Further, in the discussion section, we have revised the final paragraph to now read:

"As climate change progresses, cumulative impacts from fishing and environmental change will thus become increasingly likely and possibly more devastating than the impacts of each stressor alone⁹¹, complicating the assessment of the effectiveness of MPAs. Therefore, better understanding the complex interactions of OA with other ecosystem stressors including fishing is key for a sustainable management of Antarctic marine ecosystems under changing environmental conditions, which will likely make a larger fraction of the previously ice-covered high-latitude Southern Ocean accessible to fishing activities (Fig. 7)."

Lastly, we have included the vertical profiles of changes in temperature and oxygen in the main text of the revised manuscript instead of only including them in the supplementary material (see Fig. 5 above). We refer to these panels in the revised discussion section:

"While our model projections indicate oxygen concentrations above the typically used hypoxic threshold of 63 mmol m^{-3} ⁸² in the Antarctic MPAs for all scenarios (Supplementary Fig. 3), the projected decline in oxygen concentrations of up to $\sim 50 \text{ mmol m}^{-3}$ might nonetheless negatively impact high-latitude ecosystems in conjunction with the projected warming of up to 1°C and OA⁸³ (Fig 4)."

My final suggestion for a major improvement would be to explain more clearly what the different SSPs are and what they represent. I did not feel that this was adequately explained, in terms of the different emissions pathways and outcomes for CO₂ concentrations and warming, and what was explained was buried in the methods. Particularly if this work is to be used by people outside of the immediate field (e.g. other scientists, policy makers), I think it would really help in conveying the main messages of the paper if the SSPs were more clearly explained in the early stages of the main text.

We have added this information in the last paragraph of the introduction. It reads:

“[...] under four “shared socioeconomic pathways” (SSP) emission scenarios^{69,70}. The four SSP scenarios represent different pathways of societal development, ranging from SSP1 assuming an increasing importance of sustainable practices to the fossil-fuel intensive SSP5⁷¹. By the year 2100, the atmospheric forcing applied in this study corresponds to an increase in global 2 m air temperatures relative to the 1990s of 1.2°C (SSP1-2.6), 2.3°C (SSP2-4.5), 3.4°C (SSP3-7.0) and 4.2°C (SSP5-8.5)⁷². Atmospheric CO₂ levels in these scenarios have increased to 446 ppm, 603 ppm, 867 ppm, and 1135 ppm⁷¹ by the year 2100, respectively.”

Minor comments:

Title: I am not convinced continuance is right in this context. I suggest changing to continuation, which is more accurate.

Changed as suggested.

Line 3: ‘little is known’. Given what is known, I think this is probably an overstatement. I would suggest rewording to reflect that we do have some knowledge, but that significant gaps and uncertainties remain.

We have rephrased the sentence as follows:

“Despite a particular sensitivity to ocean acidification (OA), significant gaps remain surrounding the future carbonate chemistry of waters on the Antarctic continental shelf.”

Line 7: here you make a compelling statement on changes in hydrogen ion concentration, but this is not addressed anywhere else in the main manuscript or supplementary information to substantiate this claim. This should be incorporated and at least mentioned in the main manuscript, or removed here.

We agree with the reviewer. For this information to be placed in the abstract, it should also occur in the main text of the manuscript. In the revised manuscript, we have therefore added information on the increase in hydrogen concentration in the 2090s relative to the 1990s to Figure 3 of the revised manuscript (see Figure 9 below; this Figure was added in response to a comment by reviewer #3) and its description in the result section.

The respective part reads:

“Expressed as an increase in hydrogen ion concentration by the 2090s relative to the 1990s, the acidity on the continental shelves increases by up to 144% (top 200 m; Weddell Sea) and 130% (bottom; Weddell Sea), which is substantially more than for the whole-MPA averages (up to 131% and 80%; see numbers printed into panels in Fig. 3).”

We have also added a statement on the calculation of the increase in acidity to the revised method section:

“The change in acidity reported in the manuscript is calculated from a change in hydrogen ion concentration ($[H^+]$) between the 2090s of a given emission scenario and the 1990s of the historical simulation”

Further, for consistency with changes in the analysis (top 2000 m in the submitted paper vs. top 200 m in the revised version), we have modified the abstract:

“By 2100, we project pH declines of up to 0.36 (total scale) for the top 200 m of the water column, corresponding to a 131% increase in hydrogen ion concentration relative to the 1990s.”

Line 23-24: 'Antarctic shelf waters are a present-day sink for Canth'. I think it would be useful here to add a brief description of seasonal changes in sink/source dynamics, so that the reader has the full picture of what makes up the net annual sink.

We have added a sentence along these lines to the respective part of the introduction. It reads as follows:

"This net sink is the result of strong biologically-driven uptake in summer, while sea-ice cover typically prevents outgassing to the atmosphere in winter²⁶. Shelf waters [...]"

Line 34: East Antarctic should be East Antarctica here (because of the preceding 'in').

We thank the reviewer for spotting this. Changed as suggested.

Figure 1: I think it should be noted that the y-axis scale for the Weddell Sea is different to the others, and ideally, the ticks should include minimum and maximum bounds (rather than having any data points outside of the bounded scale).

We have revised Figure 1 in response to comments by both reviewer #1 and #3 (see Figure 1 above). In the process, we have made sure that the y-axis scale is the same for all five MPAs and that the ticks include minimum and maximum bounds.

Line 37: for the broad readership of Nature Communications, I think it is necessary to state what Phaeocystis is. Hence, I would add 'the haptophyte Phaeocystis' (or prymnesiophyte, although that may be too specific).

We have modified the sentence to read "the haptophyte *Phaeocystis*".

Line 53: should black be capitalised?

For consistency with "emerald rockcod" we have rewritten to "black rockcod" in the revised manuscript.

Line 57: there is considerable debate amongst the biological community as to whether there are any keystone species in the Southern Ocean (either Antarctic krill or silverfish), so I suggest the authors consider whether it is appropriate to use this terminology here without any substantiation. Especially the way the sentence is constructed 'keystone species, e.g. ... upper trophic level organisms' does not seem right to me without naming any organisms, so I suggest revising.

We thank the reviewer for pointing this out. In the revised manuscript, we have rephrased the respective part of the sentence to now read:

"Altogether, acknowledging knowledge gaps on direct OA impacts for important species in the Southern Ocean foodweb (e.g., Antarctic silverfish and upper-trophic level organisms), [...]"

Line 61: among with. Should be either among or with, or possibly along with

We have changed it to "along with".

Line 67-69: I suggest adding that continental shelf ecosystems are also more sensitive to the impacts of OA, to strengthen your point further.

We have modified the last sentence of the introduction, and it now reads:

"[...] resulting in more severe OA on the continental shelves, where OA-sensitive ecosystems reside^{4,42}."

Line 70-71: simulated top 2000 m average and bottom C_{anth} concentrations. It becomes clear what this means later on, but on the first mention, it is ambiguous. I suggest spelling it out more clearly on first use and then using this terminology for the rest of the paper.

We have revised the start of the result section to now read:

“Simulated C_{anth} concentrations averaged over the top 200 m of the water column are similar in magnitude on the continental shelves and in the open ocean under present-day conditions throughout the Antarctic (Fig. 2a). In contrast, future top 200 m C_{anth} concentrations and C_{anth} concentrations at the seafloor are at least two times higher on the continental shelves than in the open ocean (Fig. 2b-d; high-emission scenario SSP5-8.5).”

Please note that the start of the section has been modified further, as the analysis in Fig. 2 of the revised manuscript shows average anthropogenic carbon concentrations over the top 200m instead of over the top 2000m (see submitted manuscript).

Please note that we have adapted Figure 2 in response to a comment by both reviewer #1 and #3, and we now show the average over the top 200m to make it easier to link the changes in carbonate chemistry to biological impacts.

Figure 2: some of the text is too small to read; I suggest enlarging.

We have increased the font sizes (see Fig. 2 above).

Line 91-92: severe shelf/MPA OA. I suggest spelling this out as ‘severe OA on shelves and in the MPAs studied’ or similar, to improve readability.

Changed as suggested, and the sentence now reads:

“The strong vertical mixing on the shelves (where most of the MPA area is located, Fig. 1) is the ultimate cause of severe OA on continental shelves and in the MPAs studied.”

Line 99-100: ...up to five times stronger surface decline. This is awkward to read; I suggest rewording for clarity.

The sentence now reads:

“This is caused by a pH decline that is up to five times stronger at the surface than at 1500 m across all MPAs for all except the lowest emission scenario [...]”

Line 121-123: ‘it is projected to rise...’. Is this for all MPAs or a subset? It would help the reader’s understanding to state this explicitly here.

Agreed, we have rephrased to:

“[...] it is projected to rise to the surface in all MPAs by 2100 for the two highest-emission scenarios [...]”

Figure 5: text is too small for ice panels. Also, the label for 2100 appears to be misaligned in panel g

We have increased the font size of the legend in the sea-ice panels and aligned the label in panel g (see Figure 6 below).

Fig. 6: This is the revised version of Fig. 5 in the submitted manuscript, which will be Fig. 7 of the revised manuscript.

Line 149-151: I think it would be useful to state at the end of this sentence ‘to at least end of century’ or ‘by 2100’ or similar to really make it clear.

Agreed, rephrased as follows:

“[...] the upper ocean remains supersaturated with respect to aragonite across all MPAs until at least end of century.”

Line 153: adaptation. You could add acclimation here too.
Agreed, changed to:

“Marine organisms are highly sensitive to the rate at which OA progresses, as this determines the time span available for possible acclimation or adaptation⁷¹.”

Line 173-174: you could also point out here that this is also true for volume in the 2090s, as shown in figure 4, to further strengthen your case.
Added as suggested:

“Based on our model experiments, the volume of supersaturated water is indeed highest in this region in both the 1990s and the 2090 [...]”

Line 182-185: would it also be worth stating here that the projected increase in PP also contributes to this effect?
We have added this information to the sentence:

“[...] this can be attributed to the lower buffer capacity resulting in more oceanic CO₂ uptake per unit primary production in an acidified ocean⁷⁵ and, to a lesser extent, an overall increased primary production⁶⁹.”

Line 192: acclimatization. Is this correct or do you mean acclimation?
We thank the reviewer for spotting this, we of course meant acclimation. Corrected in the revised text.

Line 191-196: after respectively (line 196), I suggest adding a statement that this means that benthos are likely to have a smaller tolerance range to variability in conditions and also mention that this is in addition to their limited ability to move.
We have added the following statement to the text:

“The exposure to smaller pH variability implies that benthic organisms likely have a smaller tolerance range, making them more susceptible to OA given their limited ability to move.”

Line 214: have should be has (referring to reduction).
We thank the reviewer for spotting this. It was corrected in the revised text.

Line 216: ...is worrisome. Related to my major comment about strengthening wording to be more compelling, I think it would be beneficial here to use stronger wording and to elaborate on what you mean.
We have rephrased this sentence to:

“The recent reduction in sea-ice cover due to the warming of waters around the Antarctic Peninsula has already negatively affected larval abundance of the Antarctic silverfish, which uses sea ice as spawning habitat⁸⁴, suggesting that the 21st-century sea-ice retreat projected for all MPAs will negatively impact circumpolar abundances of this species (Fig. 7k-o).”

Line 230: In addition to my major comment about this paragraph, I suggest adding ‘and direct human impacts’ after ‘environmental change’.
A revised version of this sentence was added to the revised introduction:

“By preserving genetic, species and ecosystem diversity through limitation or prohibition of human activities (e.g., fishing), the implementation of MPAs can increase resilience of marine ecosystems to environmental change, including those resulting from climate change⁴⁵.”

Line 231: I suggest specifying ‘global emissions of greenhouse gases’
We have changed the text as suggested.

Line 478: <100 km. From looking at the figure, this does not appear to be true for the region between 50 and 60 S in the Pacific and Atlantic sectors. Should it be >100 or <150?
We thank the reviewer for pointing this out. The text was changed to say “<150” in the revised manuscript.

Line 512: due should be due to
We have corrected the sentence.

Line 531: state these what, so that this is not left hanging.
Changed as suggested:

“By computing simA minus simC of any model tracer or flux, we can isolate the effect of climate change on the projected changes in these tracers or fluxes.”

Line 542 (and 543, 544, 545): I suggest using standard scientific notation (x10⁶) rather than Mio.
We have changed the text as suggested by the reviewer.

Lines 544-548: It is rather confusing that the authors state that 73% of the Antarctic Peninsula MPA area is on the shelf (i.e. 27% is not) and then to say that ‘the vast majority’ of grid points are defined as shelf. This also appears to go against what is stated in the text, which is that the whole of the Peninsula area is over the shelf. It strikes me that 27% is enough to run the analysis for the open ocean component. However, I do trust the authors’ judgement on this and their approach to the analysis, so I suspect it is just a case of this wording being confusing in the methods section. Please clarify.

We thank the reviewer for this comment, as it made us realize that there was a bug in the analysis, upon which we based the decision to only show whole-MPA averages for the Antarctic Peninsula MPA. In the revised version of the manuscript, we separately show all main results for the continental shelf in this region. We have adapted all Figures accordingly (see Fig. 1, 5, 7, 9-11 in this document).

Additionally, we have modified the respective part of the method section to now read:
“As the South Orkney Islands Southern Shelf MPA is entirely defined as “open ocean” based on the model topography, we only report values for the whole MPA for this MPA in all Figures.”

References:

In general, there is very good coverage of the important references in this field and the study is placed appropriately within the wider literature. There are a small number of references that I felt were missing and would be beneficial to add for comprehensive coverage. These are:

Deppeler, S., Petrou, K., Schulz, K. G., Westwood, K., Pearce, I., Mckinlay, J., et al. (2018). Ocean acidification of a coastal Antarctic marine microbial community reveals a critical

threshold for CO₂ tolerance in phytoplankton productivity. *Biogeosciences* 15, 209–231. doi: 10.5194/bg-15-209-2018

Gutt, J., Bertler, N., Bracegirdle, T. J., Buschmann, A., Comiso, J., Hosie, G., et al. (2015). The Southern ocean ecosystem under multiple climate change stresses – an integrated circumpolar assessment. *Glob. Change Biol.* 21, 1434–1453. doi: 10.1111/gcb.12794

Hancock, A. M., Davidson, A. T., Mckinlay, J., Mckinn, A., Schulz, K. G., and Van Den Enden, R. L. (2018). Ocean acidification changes the structure of an Antarctic coastal protistan community. *Biogeosciences* 15, 2393–2410. doi: 10.5194/bg-15-2393-2018

Henley, S. F., Cavan, E. L., Fawcett, S. E., Kerr, R., Monteiro, T., Sherrell, R. M., et al. (2020) Changing Biogeochemistry of the Southern Ocean and Its Ecosystem Implications. *Front. Mar. Sci.* 7:581. doi: 10.3389/fmars.2020.00581

Tortell, P. D., Payne, C. D., Li, Y. Y., Trimborn, S., Rost, B., Smith, W. O., et al. (2008). CO₂ sensitivity of Southern Ocean phytoplankton. *Geophys. Res. Lett.* 35:L04605.

Trimborn, S., Brenneis, T., Sweet, E., and Rost, B. (2013). Sensitivity of Antarctic phytoplankton species to ocean acidification: Growth, carbon acquisition, and species interaction. *Limnol. Oceanogr.* 58, 997–1007. doi: 10.4319/lo.2013.58.3. 0997

Westwood, K. J., Thomson, P. G., Van Den Enden, R. L., Maher, L. E., Wright, S. W., and Davidson, A. T. (2018). Ocean acidification impacts primary and bacterial production in Antarctic coastal waters during austral summer. *J. Exp. Mar. Biol. Ecol.* 498, 46–60. doi: 10.1016/j.jembe.2017.11.003

We thank the reviewer for these additional references. We have included these in the revised introduction.

There are a number of places where the presentation of number ranges is grammatically incorrect. These should either be presented as ‘between x and y’, ‘from x to y’ or ‘x-y’. Examples of incorrect constructions are at lines 179 (between December-February), 487 (from 1950-2014), 556 (between 1250-3632). Please correct these.

We thank the reviewer for pointing this out. We have corrected this as suggested.

A minor grammatical point would be that there are lots of uses of an Oxford comma, which strictly speaking is incorrect and these should be removed. Examples are at lines 39 (after whales), 40 (after bivalves), 50 (after foraminifera), 490 (after heat), 534 (after input).

We have screened the text and have removed Oxford commas.

Supplementary information

Supp Figure 1: I suggest making the images larger to occupy some of the large amount of white space and to make them easier to read.

We have changed the layout of the panels to make better use of the white space (see Fig. 7 below).

Fig. 7: Supplementary Figure 1 of the revised manuscript.

Supp Figure 2: To improve readability, I suggest adding a y-axis scale to panel d to apply to panels d and e, in the same way that the scale on panel a applies to a, b and c. I would also suggest nudging the two lines of the y-axis label slightly further apart to avoid the 2s colliding. It would also help to make the labels 'annual mean' and 'DJF mean' more prominent for clarity. We thank the reviewer for these suggestions. We have implemented them all (see Fig. 8 below).

Fig. 8: Supplementary Figure 3 of the revised manuscript.

Supp Figure 3: In line 3 of the caption, profile should be profiles.
 Changed as suggested.

Supp Figure 4: In line 5 of the caption, I think cb is a typo. I also think the y axis of panel b should say calc, not arag.
 We thank the reviewer for spotting these errors. We have corrected both.

Supp Figure 5: in line 3 of the caption, the comma should be a semi-colon (;)
 Changed as suggested.

These comments and suggestions should all be amenable to being addressed by the authors, and in this case I believe the work would be ready for publication in Nature Communications. I am happy to review it again, or if the editor is happy handling this themselves, that is fine for me too.

Sian Henley
 University of Edinburgh, UK
 9/6/2023

Reviewer #3 (Remarks to the Author):

The study of Nissen and colleagues explores the implications provided by the high-resolution modelling of a much more severe OA in the Southern Ocean than reported ever before. The study excels in the modelling outputs, the scenarios are superbly linked to the physical-chemical process and produce state-of-the-art science on the ocean acidification over this century. The study really shows the importance of the model improvements and the implications it could bring.

On the other hand, the authors used the model outputs to suggest that the current predictions of the MPAs and its connectivity are not sufficient given the projected increases in OA. While this

is a solid implication of the model results, **the MPA efficiency cannot be evaluated only based on the saturation state (of 1) alone**. As the MPAs are inherently linked to the protection of marine biology, the **paper insufficiently covers the link between the chemistry and the biology**, leaving a big gap in the interpretation. So far, in the literature, there are numerous studies for variety of species and functional groups for which specific pH or omega saturation state THRESHOLDS exists. **Instead of just using a general omega =1 (which is even insufficient for the marine calcifiers impacted by dissolution), the authors have unique opportunity to apply the thresholds for different climate scenarios and show species sensitivity (or resilience)**. That would be a much stronger link to the MPAs due to climate change/OA and associated loss of habitat.

With that, I would suggest that the authors **separate the pelagic and benthic habitats** and assess the impacts separately, i.e. not only produced the change in the average over the upper 2000m, but actually **more applicable for the pelagic and benthic biota**, for example pteropods occupy the upper 200m, so project the threshold changes for that habitats, and equally the bottom for the adult benthic calcifiers. As it is now, **it is not accurate enough in terms of biological representation**. This should significantly strengthen the implications of why the MPA expansion/reconsideration is needed.

The **weakest point so far is the discussion on the MPA, implications, the only solution the enlargement along with the mitigation**. There are several other options, like fisheries management, both in catch per unit effort change and timing of fisheries, etc. I think this section needs to be scientifically beefed-up content wise, best to examine the MPAs strategies and alternatives in the best managed MPAs across the world.

We thank the reviewer for this constructive feedback. We agree that by focusing our results on averages over the top 2000m of the water column, we have made it too difficult in the submitted manuscript to link the reported changes in carbonate chemistry to biological impacts. In response to the reviewer's comment, we have changed the way in which we report our findings in the revised manuscript.

In particular, we now report changes in pH and in the saturation states (Ω) with respect to aragonite and calcite as averages over the top 200m of the water column (pelagic habitat) and in the bottom layer (benthic habitat). This change applies to Figure 1 in the submitted manuscript (see Figure 1 above, which shows the revised version). In addition, to complement the figures in the submitted manuscript, we have added time series plots of pH and Ω_{arag} to the main part of the revised paper and of Ω_{calc} to the revised supplementary material (see Figures 9-11 below). With these additions, we hope that our findings are more relevant for those studying biological impacts of ocean acidification, as we realize that different species will be impacted after different critical thresholds in pH, Ω_{arag} or Ω_{calc} have been crossed. We refer the reviewer to the third paragraph of our introduction, in which we provide examples for the sensitivities of specific species.

In the Results section, the revised or new parts read as follows:

"In the 1990s, waters on the continental shelves have a higher pH compared to the average over the regions encompassed by the whole MPA (dashed black lines on top and to the right of the solid black lines in Fig. 3 and Fig. 4, respectively). Averaged over the top 200 m, the pH in Antarctic coastal waters is up to 0.05 higher in the 1990s than in the regions encompassed by the whole MPAs (with a maximum for the East Antarctic MPA), while this difference amounts to up to 0.1 for bottom pH (Weddell Sea and East Antarctic MPA; Fig. 1). Over the 21st century, this shelf–open ocean gradient is sustained for the lower-emission scenarios, but it reverses for the two highest-emission scenarios in the regions encompassed by the Weddell Sea, East

Antarctic and Ross Sea MPAs (dashed colored lines below and to the left of the solid, colored lines in Fig. 3 and Fig. 4, respectively). By 2100, the average pH on the continental shelves is up to 0.01 (top 200 m) and 0.04 (bottom) lower than in the area of the whole MPA, again demonstrating the overall stronger acidification in the shallow continental shelf seas than in the open ocean (Fig. 1).”

“More critically, virtually all (>95%) waters in the MPAs are undersaturated with respect to aragonite for the three highest-emission scenarios, with this ubiquitous undersaturation only being avoidable for the lowest-emission scenario SSP1-2.6 (see also Fig. 6).”

“The onset of undersaturated conditions with respect to aragonite varies with depth and across seasons, scenarios and to some extent also regions (Fig. 6 & Fig. 7). Bottom waters averaged over the whole MPAs are undersaturated throughout the analysis period (on the continental shelf, only those in the Weddell Sea are initially supersaturated; second column in Fig. 6). In contrast, the saturation state in the top 200 m of the water column exceeds a value of 1 in all MPAs in the 1990s (ranging from 1.21 in the east Antarctic MPA to 1.39 in the Weddell Sea; solid lines in the first column of Fig. 6). By the 2090s, it decreases to 0.62-0.68 across regions for the highest-emission scenario and to 1.03-1.14 for the lowest-emission scenario.”

Relating to the reviewer’s summary of our main findings, we want to point out that the levels of acidification projected with our model for the *upper ocean* are in close agreement with those previously published using Earth System Models (Kwiatkowski et al., 2020). For the same high-emission scenario, differences in the projections arise for the coastal areas and especially at the subsurface, where our model simulations project acidification rates which are two times larger than those in Earth system models. We attribute these differences to the more realistic representation of dynamics on the Antarctic continental shelf in our model setup compared to most existing Earth System Models. In the revised manuscript, we have added a statement along these lines to the discussion section:

“Our model experiments, conducted with a more realistic process representation on the Antarctic continental shelf than earlier modeling studies that investigated Southern Ocean climate-change feedbacks^{2,76,77}, demonstrate that climate-change feedbacks on OA can be substantial under high-emission scenarios. While the projected upper ocean acidification under the high-emission scenario SSP5-8.5 in our model is very similar to the acidification previously reported for the high-latitude Southern Ocean in Earth system models⁷⁸, the bottom acidification rates on the Antarctic continental shelf are about two times larger in our model simulations (pH declines by more than 0.35 as opposed to by less than 0.2 in ref⁷⁸).”

We also appreciate the feedback by the reviewer on the discussion of our findings in the context of MPAs. We have revised the final paragraph of the manuscript, which now reads:

“Our 21st-century projections call for strong emission-mitigation efforts, as high-latitude Southern Ocean waters are only spared from severe OA in the lowest-emission scenario (SSP1-2.6) and as climate-change feedback substantially aggravate OA on the Antarctic continental shelf under the highest-emission scenarios (SSP5-8.5). OA across all emission scenarios within the proposed and adopted MPAs is comparable to OA in continental shelf areas not protected within an MPA (Supplementary Figure 7). This implies two things for the design and evaluation of MPAs: Firstly, as long as some portion of Antarctic waters remains outside of MPAs, these regions could serve as an important reference area for differentiating the ecosystem impacts of fishing from climate change and thus assessing the effectiveness of MPAs²⁷⁻²⁹. Secondly, given that ecosystems in high-latitude waters outside of the MPAs will be

similarly affected by severe OA, a reduction of fishing activities in a larger fraction of Antarctic coastal waters could be considered to reduce cumulative stressors on the system. In that context, our findings support the notion of today's Weddell Sea acting as a climate-change refuge²⁹, and the adoption of the Weddell Sea MPA should therefore have high priority. The active fisheries for Antarctic toothfish and Antarctic krill have been suggested to already be potentially impacting ecosystems in the Ross Sea^{88,89} and near the Antarctic Peninsula^{90,91}, respectively. As climate change progresses, cumulative impacts from fishing and environmental change will thus become increasingly likely and possibly more devastating than the impacts of each stressor alone⁹². Therefore, better understanding the complex interactions of OA with other ecosystem stressors including fishing is key for a sustainable management of Antarctic marine ecosystems under changing environmental conditions, which will likely make a larger fraction of the previously ice-covered high-latitude Southern Ocean accessible to fishing activities (Fig. 7). Altogether, the reduction of greenhouse gas emissions together with simultaneous management of fishing and other human activities, e.g., within MPAs, are necessary for safeguarding Antarctic marine ecosystems."

Fig. 9: New Figure 3 in the revised manuscript. The numbers printed in each panel denote the percent increase in acidity, which was computed from the H^+ concentration in the 2090s and the 1990s.

Fig. 10: New Figure 6 in the revised manuscript.

Fig. 11: New Supplementary Figure 2 in the revised manuscript.

Cited literature

Ainley, D. G., & Pauly, D. (2013). Fishing down the food web of the Antarctic continental shelf and slope. *Polar Record*, 50(01), 92-107. doi:10.1017/s0032247412000757

Ainley, D. G., Crockett, E. L., Eastman, J. T., Fraser, W. R., Nur, N., O'Brien, K., . . . Siniff, D. B. (2017). How overfishing a large piscine mesopredator explains growth in Ross Sea penguin populations: A

framework to better understand impacts of a controversial fishery. *Ecological Modelling*, 349, 69-75. doi:10.1016/j.ecolmodel.2016.12.021

Breitburg, D., Levin, L. A., Oschlies, A., Grégoire, M., Chavez, F. P., Conley, D. J., Garçon, V., Gilbert, D., Gutiérrez, D., Isensee, K., Jacinto, G. S., Limburg, K. E., Montes, I., Naqvi, S. W. A., Pitcher, G. C., Rabalais, N. N., Roman, M. R., Rose, K. A., Seibel, B. A., ... Zhang, J. (2018). Declining oxygen in the global ocean and coastal waters. *Science*, 359(6371). <https://doi.org/10.1126/science.aam7240>

Eayrs, C., Holland, D., Francis, D., Wagner, T., Kumar, R., & Li, X. (2019). Understanding the Seasonal Cycle of Antarctic Sea Ice Extent in the Context of Longer-Term Variability. *Reviews of Geophysics*, 57(3), 1037–1064. <https://doi.org/10.1029/2018RG000631>

Hinke, J. T., Cossio, A. M., Goebel, M. E., Reiss, C. S., Trivelpiece, W. Z., & Watters, G. M. (2017). Identifying Risk: Concurrent Overlap of the Antarctic Krill Fishery with Krill-Dependent Predators in the Scotia Sea. *PLoS One*, 12(1), e0170132. doi:10.1371/journal.pone.0170132

Kwiatkowski, L., Torres, O., Bopp, L., Aumont, O., Chamberlain, M., Christian, J. R., Dunne, J. P., Gehlen, M., Ilyina, T., John, J. G., Lenton, A., Li, H., Lovenduski, N. S., Orr, J. C., Palmieri, J., Santana-Falcón, Y., Schwinger, J., Séférian, R., Stock, C. A., ... Ziehn, T. (2020). Twenty-first century ocean warming, acidification, deoxygenation, and upper-ocean nutrient and primary production decline from CMIP6 model projections. *Biogeosciences*, 17(13), 3439–3470. <https://doi.org/10.5194/bg-17-3439-2020>

Roberts, C. M., O'Leary, B. C., McCauley, D. J., Cury, P. M., Duarte, C. M., Lubchenco, J., Pauly, D., Sáenz-Arroyo, A., Sumaila, U. R., Wilson, R. W., Worm, B., & Castilla, J. C. (2017). Marine reserves can mitigate and promote adaptation to climate change. *Proceedings of the National Academy of Sciences*, 114(24), 6167–6175. <https://doi.org/10.1073/pnas.1701262114>

Ryan, C., Santangelo, M., Stephenson, B., Wilson, E., & Savoca, M. (2023). Commercial krill fishing within a foraging supergroup of fin whales in the Southern Ocean. *Ecology*, 104(4), e4002.

Salas, L., Nur, N., Ainley, D. G., Burns, J., Rotella, J., & Ballard, G. (2017). Coping with the loss of large, energy-dense prey: a potential bottleneck for Weddell Seals in the Ross Sea. *Ecological Applications*, 27(1), 10-25.

Steinberg, D. K., & Landry, M. R. (2017). Zooplankton and the Ocean Carbon Cycle. *Annual Review of Marine Science*, 9(1), 413–444. <https://doi.org/10.1146/annurev-marine-010814-015924>

Teschke, K., Brtnik, P., Hain, S., Herata, H., Liebschner, A., Pehlke, H., & Brey, T. (2021). Planning marine protected areas under the CCAMLR regime – The case of the Weddell Sea (Antarctica). *Marine Policy*, 124(December 2020), 104370. <https://doi.org/10.1016/j.marpol.2020.104370>

Trathan, P. N., Warwick-Evans, V., Young, E. F., Friedlaender, A., Kim, J. H., & Kokubun, N. (2022). The ecosystem approach to management of the Antarctic krill fishery - the 'devils are in the detail' at small spatial and temporal scales. *Journal of Marine Systems*, 225. doi:10.1016/j.jmarsys.2021.103598

Trivelpiece, W. Z., Hinke, J. T., Miller, A. K., Reiss, C. S., Trivelpiece, S. G., & Watters, G. M. (2011). Variability in krill biomass links harvesting and climate warming to penguin population changes in Antarctica. *Proceedings of the National Academy of Science*, 108(18), 7625-7628. doi:10.1073/pnas.1016560108

REVIEWER COMMENTS

Reviewer #1 (Remarks to the Author):

The authors have provided an excellent and detailed response to the first reviews. As clearly articulated in the response reviews and in the revised manuscript, the authors have addressed my original comments and I recommend publishing the paper. I have no further comments.

Reviewer #2 (Remarks to the Author):

Nissen et al. have addressed all of my comments from the first round of review in a satisfactory manner. The manuscript text has changed quite substantially, as have some of the figures. This further work has certainly led to improvements in the quality of the manuscript. In my opinion, this manuscript will be acceptable for publication in Nature Communications once some matters arising from this resubmitted version have been addressed.

The only moderate-to-major comment would be that some of the new sections do not seem as well polished and streamlined as they might be. This is particularly the case for the discussion and conclusions, where I thought the wording could be tighter overall and this would help in maximising the impactful delivery of the work done. I would recommend that the authors go back through this section and the other new sections and make sure the writing is as clear, concise and compelling as it can be. There is nothing wrong, as such, with it at present; I just feel like some of the text could be better polished and streamlined to really make the most of these important findings and considerations.

In addition, I have the following minor points to be addressed.

Line 82: I suggest "... will have increased..."

Line 92: I suggest bringing "[in the] high-emission scenario SSP5-8.5" out of the brackets as this is an important part of the sentence

Line 108: there is inconsistency as to whether you name this MPA as South Orkney Southern Shelf or South Orkney Islands Southern Shelf. Ensure that this is correct and consistent throughout. Note: I think the shortening to South Orkney MPA later in the text is acceptable, once it has been defined properly earlier on.

Line 131: 'is altered' should be 'are altered' (for magnitude and sign).

Line 149: should this be upper 200 m, given the changes to the analysis since the original submission? If so, this should be changed on the figure and in its caption as well. I'm not sure which is right in this instance, but check. Check this for figure S7 as well.

Lines 219-224: it would be useful to identify which scenarios you are talking about somewhere here. I assume SSP5-8.5 and SSP3-7.0, but it would be worth being explicit.

Line 245-246: should be Supplementary Figure 4

Line 247: "is possibly non-negligible" seems very vague and rather clunky. I would suggest "could also be a factor" or "cannot be ruled out" for improved clarity.

Line 248: should be Supplementary Figure 5

Line 259: I would suggest finishing this sentence with "in the Southern Ocean" for completeness.

Line 268: "of" not required - I suggest deleting.

Line 276-279: it would be useful here to state "at all depths" or similar (if this is true), or to define the depth range(s) you are referring to.

Line 279-282: I suggest rewording to "Even though it remains unclear in laboratory experiments to what extent...". Also, for completeness, I would add explicit mention of impacts to their prey and/or predators (and hence how organisms not directly impacted are affected by indirect effects).

Line 301: climate-change feedbacks?

Line 302: should be "highest-emission scenario"

Line 318: "for sustainable management" or "for the sustainable management"

Figure S2: following my comments in the first round of reviews, I think line 2 of this caption should read "...to calcite", consistent with the omega-calc in brackets

I believe that all of these comments and suggestions are amenable to being addressed, and once this has been done, the manuscript will be ready for publication. I am happy to review the second round of revisions if required, or for the editor to sign off if they feel my comments in this round have been addressed satisfactorily.

Sian Henley
University of Edinburgh
20/10/2023

Reviewer #3 (Remarks to the Author):

The revised manuscript by Nissen et al. improved the quality of the interpretation with respect to the projected changes over the shelf and MPAs areas. Overall, the modelling results are excellent and could and should serve to make future decisions on the MPAs in the Southern Ocean.

However, the study with the latest improvements, is severely unbalanced in text wrt to MPA framing. Most of the text deals with the modelling outputs and only the last page starts discussing the implications for the MPA, while the title and the abstract point out this to be major force behind this MS. While the MS describes the modelling efforts in detail, it falls short of what to do with them, the implications are too weak, too vague and inadequate. I strongly suggest restructuring, the MPAs text needs to be extended, more precise- like where are suitable areas, where should that continuous expansion be and over what extent. Otherwise, such MS cannot really be used for policy informed discussions, with high relevance for the Nature.

In general, seeing more nuanced results of this study now, I have serious reservations about the current thinking about expanding MPAs. As this study finds out, MPAs would see the greatest OA severity precisely in the MPAs. The decision to expand the areas would contradict the common sense of extending the areas that would be most compromised. In contrast, MPAs should be positioned in the areas with the greatest protection that can act as a refuge. Simply allowing for only a reference area (line 305) that authors cite is merely not enough. The information from the modelling should point out where the areas of potential refuge are the greatest (including the projections and feedback) and then build the case on this. As it currently is, I simply see such justification to be too weak.

Upon building the case of the greatest refuge, this also means that the areas need to be looked at more comprehensively, not evaluating only OA, but also warming and hypoxia, because intense warming could momentarily restructure the ecosystem. I find it inadequate to propose the expansion of MPAs without taking and evaluating the extent of other stressors. I suggest the authors to delineate the projected changes in warming, hypoxia and OA for the three investigated areas (Weddel, East Ant, West Ant Peninsula), aligned them together and find most suitable refuge areas for the same functional groups already considered. I think the hypoxia threshold currently used is OK, it might be the most early-warning but is sufficient. As for thermal reference, a lot of these thresholds exist in the literature. Once all three stressors are considered, a more precise advice on where the expansion of the MPAs should be- that would be the most comprehensive advice that the CAMLR needs to make the next steps.

Minor comments:

Line 47-49: Need to be some quantitative citation of MPA protection, increased in the diversity protection etc.

Demonstrate high biodiversity in the MPAs with more references -line 260

Ref 62, 63, 60, 61, 67

Line 186 several years of adaptation time?

Immobility – line 263- that is only true for the benthic organisms, expand the text for this to be relevant

Line 309 – at least some evaluation what the reduction of fishing could do

Reviewer #1 (Remarks to the Author):

The authors have provided an excellent and detailed response to the first reviews. As clearly articulated in the response reviews and in the revised manuscript, the authors have addressed my original comments and I recommend publishing the paper. I have no further comments.

We thank reviewer #1 for the positive feedback.

Reviewer #2 (Remarks to the Author):

Nissen et al. have addressed all of my comments from the first round of review in a satisfactory manner. The manuscript text has changed quite substantially, as have some of the figures. This further work has certainly led to improvements in the quality of the manuscript. In my opinion, this manuscript will be acceptable for publication in Nature Communications once some matters arising from this resubmitted version have been addressed.

The only moderate-to-major comment would be that some of the new sections do not seem as well polished and streamlined as they might be. This is particularly the case for the discussion and conclusions, where I thought the wording could be tighter overall and this would help maximising the impactful delivery of the work done. I would recommend that the authors go back through this section and the other new sections and make sure the writing is as clear, concise and compelling as it can be. There is nothing wrong, as such, with it at present; I just feel like some of the text could be better polished and streamlined to really make the most of these important findings and considerations.

We thank Sian Henley for carefully reading our revised manuscript. We have revised the text following the suggestions below. Further, we made some additional minor modifications in the discussion section in an attempt to streamline the text (please see track-change document).

In addition, I have the following minor points to be addressed.

Line 82: I suggest "... will have increased..."

Changed as suggested.

Line 92: I suggest bringing "[in the] high-emission scenario SSP5-8.5" out of the brackets as this is an important part of the sentence

Changed as suggested.

Line 108: there is inconsistency as to whether you name this MPA as South Orkney Southern Shelf or South Orkney Islands Southern Shelf. Ensure that this is correct and consistent throughout. Note: I think the shortening to South Orkney MPA later in the text is acceptable, once it has been defined properly earlier on.

We thank the reviewer for pointing out this inconsistency. In the revised manuscript, we now consistently refer to this MPA as the "South Orkney Islands Southern Shelf MPA". Similarly, we have also ensured consistent naming of the second adopted MPA, i.e., the "Ross Sea *region* MPA".

Line 131: 'is altered' should be 'are altered' (for magnitude and sign).

Changed as suggested.

Line 149: should this be upper 200 m, given the changes to the analysis since the original submission? If so, this should be changed on the figure and in its caption as well. I'm not sure which is right in this instance, but check. Check this for figure S7 as well.

We thank the reviewer for paying such close attention. We confirm that what is currently stated in the text, figure, and figure caption is correct: This analysis still covers the upper 2000m of the water column, as we think that the division into volumes with a specific omega range is more meaningful when applied to a larger part of the water column/habitat than when only applying this division to the top 200m.

Lines 219-224: it would be useful to identify which scenarios you are talking about somewhere here. I assume SSP5-8.5 and SSP3-7.0, but it would be worth being explicit.

We are here referring to the emission scenarios SSP2-4.5, SSP3-7.0 and SSP5-8.5. We have clarified this as follows in the revised text:

“Under the intermediate to high emission scenarios SSP2-4.5, SSP3-7.0 and SSP5-8.5, ecosystems in the shallow continental shelf seas of the proposed and adopted Antarctic MPAs will be exposed to severe OA by 2100. For the SSP3-7.0 and SSP5-8.5 scenarios, aragonite undersaturation is ubiquitous [...].”

Line 245-246: should be Supplementary Figure 4

Changed as suggested.

Line 247: "is possibly non-negligible" seems very vague and rather clunky. I would suggest "could also be a factor" or "cannot be ruled out" for improved clarity.

We have rephrased as suggested to:

“[...] the contribution of changes in CO₂ solubility and of a redistribution of water masses could also be a factor [...].”

Line 248: should be Supplementary Figure 5

Changed as suggested.

Line 259: I would suggest finishing this sentence with "in the Southern Ocean" for completeness.

Changed as suggested.

Line 268: "of" not required - I suggest deleting.

Changed as suggested.

Line 276-279: it would be useful here to state "at all depths" or similar (if this is true), or to define the depth range(s) you are referring to.

We have revised this sentence as follows:

"However, in the two highest-emission scenarios, the projected pH by 2100 is <7.8 throughout the water column for all MPAs (Fig. 1) [...]"

Line 279-282: I suggest rewording to "Even though it remains unclear in laboratory experiments to what extent...". Also, for completeness, I would add explicit mention of impacts to their prey and/or predators (and hence how organisms not directly impacted are affected by indirect effects).

We have rephrased the sentence to now read:

"Even though it remains unclear in laboratory experiments to what extent a longer acclimation of adult organisms can reduce the severe impacts reported for egg development and juveniles², strong OA would almost certainly disrupt food webs, as OA might affect even those organisms which are not directly negatively impacted by OA themselves through direct OA effects on their prey or predators."

Line 301: climate-change feedbacks?

Changed as suggested.

Line 302: should be "highest-emission scenario"

Changed as suggested.

Line 318: "for sustainable management" or "for the sustainable management"

Changed to "for sustainable management".

Figure S2: following my comments in the first round of reviews, I think line 2 of this caption should read "...to calcite", consistent with the omega-calc in brackets

This has been corrected in the Figure caption.

I believe that all of these comments and suggestions are amenable to being addressed, and once this has been done, the manuscript will be ready for publication. I am happy to review the second round of revisions if required, or for the editor to sign off if they feel my comments in this round have been addressed satisfactorily.

Sian Henley
University of Edinburgh
20/10/2023

Reviewer #3 (Remarks to the Author):

The revised manuscript by Nissen et al. improved the quality of the interpretation with respect to the projected changes over the shelf and MPAs areas. Overall, the modelling results are excellent and could and should serve to make future decisions on the MPAs in the Southern Ocean.

However, the study with the latest improvements, is severely unbalanced in text wrt to MPA framing. Most of the text deals with the modelling outputs and only the last page starts discussing the implications for the MPA, while the title and the abstract point out this to be major force behind this MS. While the MS describes the modelling efforts in detail, it falls short of what to do with them, the implications are too weak, too vague and inadequate. I strongly suggest restructuring, the MPAs text needs to be extended, more precise- like where are suitable areas, where should that continuous expansion be and over what extent. Otherwise, such MS cannot really be used for policy informed discussions, with high relevance for the Nature.

In general, seeing more nuanced results of this study now, I have serious reservations about the current thinking about expanding MPAs. As this study finds out, MPAs would see the greatest OA severity precisely in the MPAs. The decision to expand the areas would contradict the common sense of extending the areas that would be most compromised. In contrast, MPAs should be positioned in the areas with the greatest protection that can act as a refugee. Simply allowing for only a refuge area (line 305) that authors cite is merely not enough. The information from the modelling should point out where the areas of potential refugee are the greatest (including the projections and feedback) and then build the case on this. As it currently is, I simply see such justification to be too weak.

Upon building the case of the greatest refugee, this also means that the areas need to be looked at more comprehensively, not evaluating only OA, but also warming and hypoxia, because intense warming could momentarily restructure the ecosystem. I find it inadequate to propose the expansion of MPAs without taking and evaluating the extent of other stressors. I suggest the authors to delineate the projected changes in warming, hypoxia and OA for the three investigated areas (Weddel, East Ant, West Ant Peninsula), aligned them together and find most suitable refugee areas for the same functional groups already considered. I think the hypoxia threshold currently used is OK, it might be the most early-warning but is sufficient. As for thermal reference, a lot of these thresholds exist in the literature. Once all three stressors are considered, a more precise advice on where the expansion of the MPAs should be- that would be the most comprehensive advice that the CAMLR needs to make the next steps.

We appreciate the reviewer's feedback, however, much of what is suggested is very much beyond the scope of this paper. Below, we seek to clarify the goal and scope of our study and its implications.

The goal of our study is to present novel future projections of ocean acidification within the bounds of existing (adopted and proposed) Antarctic MPAs. As such, most of the manuscript text focuses on presenting the modeling results in these predefined regions. Serving as a motivation for our study and providing the analysis framework, MPAs are prominently mentioned in the abstract and introduction of the manuscript, and our findings on ocean acidification are placed into the context of MPA policymaking in the discussion section. We thus believe the title of our manuscript adequately summarizes its content. After the first round of reviews, we have made major modifications to the manuscript. In particular, in response to comments by reviewer #1, who is happy with the revised manuscript, we decided to put less emphasis on making recommendations for MPAs in the revised manuscript and instead highlight our modeling results more. With our manuscript, we do not endeavor to propose modifications to existing MPA boundaries -- these have already been established by means of bioregionalization and political compromises (Brooks et al., 2020, 2021). We emphasize that priorities for protection, and resultant boundaries of the MPAs, are a policy decision, and ultimately a political process, which

can be supported by science. Here we seek to build on and provide that science which could be utilized in the policy process.

We agree with the reviewer that a comprehensive assessment of Antarctic MPA boundaries is needed for the CCAMLR community, especially one which considers the present-day distributions and future projections of all ecosystem stressors including acidification. However, such an assessment is well beyond the scope of this manuscript, which focuses on projections of acidification within the existing (adopted and proposed) MPA boundaries. To address this reviewer point, we have added a sentence in the discussion section to clarify future research priorities:

“Future work should include a joint assessment of the evolution of all ecosystem stressors in Antarctic coastal waters over the 21st century to identify regions of greatest and least risk for marine organisms.”

Finally, we believe that MPA design and implementation should indeed consider regions characterized by rapid change, as a reduction in other stressors (e.g., fishing) in these regions will contribute to reducing the cumulative threat to ecosystems.

Minor comments:

Line 47-49: Need to be some quantitative citation of MPA protection, increased in the diversity protection etc.

In the revised manuscript, we have added two more citations (Lester et al., 2009; Edgar et al., 2014), which quantify the positive impact of MPAs on marine ecosystems, e.g., in preserving biodiversity:

“[...] the implementation of MPAs can increase resilience of marine ecosystems to environmental change, including those resulting from climate change⁴⁷⁻⁴⁹.”

Demonstrate high biodiversity in the MPAs with mor references -line 260
Ref 62, 63, 60, 61, 67

We have added two references to the revised manuscript (De Broyer et al., 2014; Douglass et al., 2014) illustrating the high biodiversity in the Southern Ocean MPAs. We cite these new references both in the revised introduction and the revised discussion section:

“Designed to protect the unique high-latitude Southern Ocean biodiversity^{27,28}, [...]”

“Given the high biodiversity in MPAs^{27,28}, [...]”

Line 186 several years of adaptation time?

Yes. Compared to organisms with a short generation time of days or weeks, e.g., phytoplankton, all organisms with a substantially longer generation time of years to decades will require a much longer time to successfully adapt to OA. For these organisms, a lower rate of OA will thus be especially beneficial. Further, substantial adaptation to OA in these organisms could only occur over a single generation if enough genetic variability is already present in the population, i.e., if no new mutations are necessary (Kelly et al., 2013). But, depending on the generation time, this would still take several years at least.

Immobility – line 263- that is only true for the benthic organisms, expand the text for this to be relevant

With other organisms being listed in the previous sentence, the purpose of this sentence is to explain why the impacts on benthic organisms can be expected to be substantial. We therefore decided to leave this sentence unchanged, but instead restate the expected negative impacts on the other listed organism groups in the preceding sentence, which now reads:

“Given the high biodiversity in MPAs, the projected severe OA under higher-emission scenarios will likely impact organisms on many trophic levels, ranging from primary producers (growth)⁵¹ to fish (metabolic capacity and mortality)^{3,66} and calcium carbonate-forming benthic organisms (shell dissolution; Fig. 1)^{4,68}.”

Line 309 – at least some evaluation what the reduction of fishing could do

To link this statement in the discussion back to what was already stated in the introduction, we have changed this part of the sentence to now read:

“[...] a reduction of fishing activities in a larger fraction of Antarctic coastal waters could be considered to reduce cumulative stressors on the system and to preserve genetic, species and ecosystem diversity⁴⁷⁻⁴⁹ [...]”

Further, we have included citations to references demonstrating the positive impacts of reduced fishing pressure (see our answer above to the minor comment on L. 47-49).

Cited literature:

Brooks, C. M., Bloom, E., Kavanagh, A., Nocito, E. S., Watters, G. M., & Weller, J. (2021). The Ross Sea, Antarctica: A highly protected MPA in international waters. *Marine Policy*, 134, 104795.

<https://doi.org/10.1016/j.marpol.2021.104795>

Brooks, C. M., Chown, S. L., Douglass, L. L., Raymond, B. P., Shaw, J. D., Sylvester, Z. T., & Torrens, C. L. (2020). Progress towards a representative network of Southern Ocean protected areas. *PLOS ONE*, 15(4), e0231361. <https://doi.org/10.1371/journal.pone.0231361>

Douglass, L. L., Turner, J., Grantham, H. S., Kaiser, S., Constable, A., Nicoll, R., Raymond, B., Post, A., Brandt, A., & Beaver, D. (2014). A Hierarchical Classification of Benthic Biodiversity and Assessment of Protected Areas in the Southern Ocean. *PLoS ONE*, 9(7), e100551.

<https://doi.org/10.1371/journal.pone.0100551>

De Broyer, C., Koubbi, P., Griffiths, H.J., Raymond, B., Udekem d'Acoz, C. d', Van de Putte, A.P., Danis, B., David, B., Grant, S., Gutt, J., Held, C., Hosie, G., Huettmann, F., Post, A., Ropert-Coudert, Y. (eds.): *Biogeographic Atlas of the Southern Ocean*. 510 pp., 2014. Cambridge, SCAR. ISBN 978-0-948277-28-3

Edgar, G. J., Stuart-Smith, R. D., Willis, T. J., Kininmonth, S., Baker, S. C., Banks, S., Barrett, N. S., Becerro, M. A., Bernard, A. T. F., Berkhout, J., Buxton, C. D., Campbell, S. J., Cooper, A. T., Davey, M., Edgar, S. C., Försterra, G., Galván, D. E., Irigoyen, A. J., Kushner, D. J., ... Thomson, R. J. (2014). Global conservation outcomes depend on marine protected areas with five key features. *Nature*, 506(7487), 216–220. <https://doi.org/10.1038/nature13022>

Kelly, M. W., & Hofmann, G. E. (2013). Adaptation and the physiology of ocean acidification. *Functional Ecology*, 27(4), 980–990. <https://doi.org/10.1111/j.1365-2435.2012.02061.x>

Lester, S. E., Halpern, B. S., Grorud-Colvert, K., Lubchenco, J., Ruttenberg, B. I., Gaines, S. D., Aíramé, S., & Warner, R. R. (2009). Biological effects within no-take marine reserves: A global synthesis. *Marine Ecology Progress Series*, 384, 33–46. <https://doi.org/10.3354/meps08029>